# FRUGALRAG: LESS IS MORE IN RL FINETUNING FOR MULTI-HOP QUESTION ANSWERING

**Abhinav Java, Srivathsan Koundinyan, Nagarajan Natarajan & Amit Sharma**
Microsoft Research India

## ABSTRACT

Reinforcement learning (RL) based on the final answer's reward has driven recent progress in small language models (SLMs) on reasoning-heavy tasks such as math and code. However, applying the same techniques to retrieval-augmented generation (RAG) benchmarks like multi-hop QA has yielded limited gains—often trailing supervised or prompting-only baselines. Instead, we argue that a viable path for RL in multi-hop QA is to use test-time scaling judiciously, for optimizing both the final answer accuracy and the efficiency in reaching that answer. We propose FrugalRAG, a two-stage finetuning framework that adaptively reduces the number of retrieval steps based on a question's difficulty. First, we train an SLM with supervised finetuning on a full-exploration policy that generates broad sub-queries. Then, we apply RL to adaptively prune search depth based on question difficulty, directly rewarding policies that balance correctness with frugality. Unlike prior approaches requiring 100× more data, our method achieves competitive performance with only 1,000 examples. On HotPotQA and other multi-hop QA benchmarks, FrugalRAG attains state-of-the-art efficiency–accuracy tradeoffs, cutting retrieval cost nearly in half. Moreover, on the challenging BrowseCompPlus benchmark, it generalizes zero-shot and surpasses SLM-based and other baselines. These results demonstrate the use of RL—not to increase reasoning steps but to optimize them—as an effective solution for scalable, efficient RAG.

## 1 INTRODUCTION

We study the problem of answering questions, such as "*Can a microwave melt Toyota Prius battery?*", given access to a large unstructured corpus like Wikipedia. The de facto approach to solving the problem is to use language models (LMs) coupled with the ability to retrieve relevant documents (e.g., Wiki passages) against queries, i.e., the retrieval-augmented generation (RAG) paradigm (Jeong et al., 2024; Jiang et al., 2023; Chan et al., 2024; Asai et al., 2023). However, answering complex questions often requires multi-hop reasoning and retrieval, i.e., the LM has to iteratively decompose the user utterance into sub-queries or search phrases (e.g., "*melting point of Toyota Prius battery*"), retrieve documents relevant to the sub-queries, and reason through the retrieved documents to issue further sub-queries, until the LM is able to generate an answer for the original query.

Based on the success of reinforcement learning (RL) in math and code applications (DeepSeek-AI, 2025), recent work applies RL-based finetuning to optimize retriever tool calling and generation of the right queries to answer a given question. However, the accuracy gains in multi-hop QA benchmarks are less impressive. For instance, on the HotPotQA benchmark (Yang et al., 2018), a simple ReAct-based Yao et al. (2023) strategy of letting a model generate up to 10 subqueries leads to higher document recall (63%) than state-of-the-art RL methods such as Search-R1 (see Table 2). Therefore, the key question is not how RL can be used to improve accuracy, rather how RL can be used to optimize the search process and make a RAG system more efficient (e.g., for HotPotQA, most questions can ideally be solved in 2-3 search queries). Ideally, we want a system that can adapt the number of search queries based on a question's difficulty level.

On efficiency, a second problem is the availability of labelled training data that maps a question to its correct answer. Outside of general web question-answering, it may be difficult to obtain ground-truth labels for questions when adapting a RAG system to work in a real-world application domain (with potentially private documents). Typical solutions in the literature (Jin et al., 2025a; Hsu et al., 2024),

however, train on 90,000-1,00,000 examples from existing benchmarks such as HotPotQA (Yang et al., 2018). Therefore, a second key problem is finetuning a language model for retrieval when labelled data is scarce. For concreteness, we set the number of training examples as 1000, an order of magnitude lower than existing efforts.

Under these considerations, we ask the question: *How can we train a model to answer questions using as few search calls as necessary? And can we do it using only 1000 training examples?* Our solution, **FRUGALRAG**, is a two-stage framework that uses exploration to generate a large number of search queries per question (Stage 1), and uses RL to optimize the number of search queries per question (stage 2). Specifically, in the *first stage*, the model is trained to maximize evidence coverage by generating diverse and informative search queries across multiple hops. In the *second stage*, we post-train the model using RL to decide when to stop retrieving and generate an answer. This decision is modeled explicitly, allowing the model to weigh the cost of further retrievals against the confidence in the retrieved evidence. Optimizing for coverage and efficiency in a single stage leads to unstable training; we find that models either over-retrieve or stop too early. Our key insight is that learning when to stop is more naturally learned through reinforcement learning (RL) signals, whereas better coverage can be obtained by repeatedly issuing high quality search queries using frameworks such as ReAct.

We evaluate FRUGALRAG on standard multi-hop datasets such as HotPotQA (Yang et al., 2018), 2WikiMultiHopQA (Ho et al., 2020) and MuSiQue (Trivedi et al., 2022b), using both document retrieval metrics such as recall and answer quality metrics. Compared to baselines, we show that FRUGALRAG obtains the highest document recall and answer quality while incurring a low number of search queries per question. In particular, our models obtain state-of-the-art accuracy on the recently released FlashRAG index benchmark for HotPotQA and MuSiQue, despite being trained on only 1000 examples.

We also find that the same FRUGALRAG model generalizes to harder information search tasks. A recently released benchmark from OpenAI tests models' abilities on challenging needle-in-a-haystack problems, wherein the reasoning steps (and hence the search queries) to find the answer can be much larger. The same FRUGALRAG 7B model described above generalizes zero-shot on this out-of-domain task; obtaining an accuracy of 20%, which is higher than that of bigger models such as DeepSeek-R1 and Search-R1-32B. Remarkably, it adapts to issue a larger number of queries for the BrowseComp data compared to earlier datasets, unlike other RL methods such as Search-R1 (Jin et al., 2025a).

## 2 RELATED WORK

In this section, we present a detailed overview of the relevant literature comprising representative multi-hop RAG methods, recent work leveraging reinforcement learning for search, and finally standard metrics for evaluation.

**Multi-Hop RAG.** Retrieval Augmented Generation is an active area of research that aims to ground the responses generated by LLMs in real world information, tackling fundamental limitations like hallucinations and trustworthiness. Several works (Guu et al., 2020; Lewis et al., 2021; Shao et al., 2023; Jeong et al., 2024; Jiang et al., 2023; Asai et al., 2023; Chan et al., 2024; Hsu et al., 2024; Li et al., 2025; Wang et al., 2025) have been proposed in recent literature that explore this problem. Guu et al. (2020) and Lewis et al. (2021) showed that augmenting the inputs of language models with retrieved information significantly improved their performance on knowledge intensive tasks like question answering. Subsequently, Trivedi et al. (2022a) leveraged few-shot Chain-of-Thought (Wei et al., 2022) prompts and iteratively retrieved information for complex questions that required multiple reasoning and retrieval steps (i.e. multi-hop settings). Shao et al. (2023) also incorporate a similar iterative retrieval approach. By using the intermediate traces, the IRCoT is able to decide what to retrieve by issuing the right search queries. Iter-RetGen (Shao et al., 2023) improves evidence gathering in multi-hop scenarios by combining retrieval and generation iteratively, such that a model's response is incorporated in the reasoning trace. However, both IRCoT (Trivedi et al., 2022a) and Iter-RetGen (Shao et al., 2023) rely on a fixed or predefined number of retrieval loops at inference, offering limited control over latency. Toolformer(Schick et al., 2023) uses a self-supervised objective to train an external model that decides to call tools (like Bing and Google search engines). ReAct (Yao et al., 2023) uses a general prompting framework by interleaving thoughts, actions, and tool calls

in order to perform complex tasks. Khattab et al. (2023) invoked LLMs through declarative calls, allowing a more structured programmatic way of calling LLMs. Further Khattab et al. (2023) also design example-data efficient compilers to optimize a given metric. Ammann et al. (2025) introduce a question-decomposition and reranking system for multi-hop RAG. Their approach improves retrieval quality through decomposition, whereas our approach focuses primarily on learning to adaptively retrieve. Collab-RAG (Xu et al., 2025), proposes collaboration between an LLM and an SLM. This method relies on coordination between models, whereas FRUGALRAG operates only with an SLM.

Beyond prompting, SelfRAG (Asai et al., 2023) trains small language models (SLMs) with dense supervision from stronger models to decide when to retrieve external information during question answering. RQRAG (Chan et al., 2024) leverages LLMs to curate datasets that teach smaller models essential skills such as query decomposition and backtracking. In addition, Chan et al. (2024) highlight the effectiveness of advanced decoding strategies such as tree-of-thought sampling for identifying high-quality reasoning trajectories. More recently, approaches like CoRAG (Wang et al., 2025) have scaled test-time computation by jointly reasoning over multiple trajectories and training models through rejection sampling. However, these methods (SelfRAG, RQRAG, CoRAG) demand substantial supervision (100k+ annotated examples) and are constrained either by inference inefficiency (RQRAG (Chan et al., 2024)) or by limited flexibility (CoRAG (Wang et al., 2025)), which relies on predefined number of search). Other works, such as DRAGIN (Su et al., 2024) and Adaptive RAG (Jeong et al., 2024), train multiple modules with large-scale supervised data to enable dynamic retrieval. SimpleDeepResearcher (Sun et al., 2025) propose data-engineering strategies to mitigate the problem of domain shift between training and testing environment showing that supervised finetuning on a small number of representative samples can yield strong improvements. In contrast to these approaches, we introduce the first method that jointly accounts for both training and inference costs. Our key intuition is that reinforcement learning can be leveraged to scale down search effectively, thereby reducing supervision requirements while maintaining competitive performance.

**Reinforcement Learning for Search.** Recently, framing search query as an RL problem has received attention. LeReT (Hsu et al., 2024) is an early work that adopted RL by performing preference optimization using diverse few shot prompts leveraging hundred-thousands of ground truth annotated documents. However, this finetuning process is computationally expensive and cannot be cheaply and readily generalized to variable-hop scenarios. As a result, LeReT utilizes a fixed amount of compute (i.e. search calls) per instance during inference. Similarly, concurrent works, Jin et al. (2025a) and Chen et al. (2025a) propose end-to-end RL-based optimization that only leverages the final answer annotation. R1-Seacher (Song et al., 2025) proposes an RL-only strategy for finetuning demonstrating strong performance on multi-hop RAG benchmarks by utilizing a curated balanced dataset with outcome based rewards. These methods show that RL can effectively be used to teach the search query generator model *to issue more search queries* for multi-hop problems *without considering latency*. Our two-stage RL framework, by contrast, first explores without RL to maximize recall and then learns to stop at test time using RL.

**Metrics for Evaluation.** Multi-hop QA involves two sub-tasks: retrieving relevant documents, and then answering the question based on the documents. Some methods report document retrieval-specific metrics such as recall (Hsu et al., 2024) whereas others report final answer metrics such as exact match (Jin et al., 2025a). Typically, a model is trained to optimize a particular metric (such as recall) and also evaluated on the same metric. For robustness, in this work we train on the recall metric and test on all metrics, including a robust model based evaluation of the final answer following FlashRAG (Jin et al., 2025b). Unlike Wang et al. (2025); Li et al. (2025); Jin et al. (2025a); Song et al. (2025) we train a reasoner module which performs reasoning and issues search calls and use an off-the-shelf final answer generator.

## 3 FRUGALRAG: EFFICIENT TWO-STAGE FINETUNING FOR RAG

We describe FRUGALRAG, a novel framework for enhancing retrieval augmented generation in LMs by decoupling the evidence exploration stage from answer generation. FRUGALRAG demonstrates several key advantages over contemporary RAG approaches – (1) *Requires only 1000 annotated training examples* which is a 100 times reduction in dataset size compared to existing works (Jin et al., 2025a; Hsu et al., 2024; Chan et al., 2024), (2) *Dynamically adapts test-time compute* which results

in low inference time latency and high retrieval recall, unlike existing fixed compute methods (Hsu et al., 2024). Algorithm 1 summarizes our framework.

**Problem Formulation.** Let $Q$ denote a complex user question that requires multiple iterative retrievals to answer. Let $f$ denote a "reasoner" language model (LM) that, at each reasoning hop, examines the current context and determines the next action. At hop $h$ (where $1 \leq h \leq B$, and $B$ is the maximum allowed number of hops), the model $f$ produces a *thought–action-search query* triplet $(T_h, A_h, S_h)$. The search query $S_h$ is passed to a retriever $\mathcal{R}(\cdot)$, which returns a set of documents: $\mathcal{D}_h = \mathcal{R}(S_h)$ from the document index $\mathcal{I}$.

Let $\mathcal{D}_0$ denote the initial context which is either empty or initialized as $\mathcal{R}(Q)$. At hop $h$, the model has access to the context: $\{Q\} \cup \{(\mathcal{D}_h, T_h, A_h, S_h)\}_0^{h-1}$. This includes the original query, all previously retrieved documents, and the previously generated thought, action, search query triplets. The process continues until the model outputs a special FINISH action, terminating after $h_{\text{term}}$ hops (or at the budget limit $B$). At this point, a separate generator LM $g$ is invoked to produce the final answer, conditioned on the original question $Q$ and the full context accumulated up to termination.

> *The central challenge* for $f$ is to iteratively construct highly targeted queries $S_h$ such that the retriever $\mathcal{R}$ can surface a minimal, sufficient set of documents to answer $Q$ within $B$ hops.

*Our key observation* is that, with sufficient test-time compute, even base models can generate multiple, diverse search queries to address a question (e.g., see Sec 4, Table 2, ReAct+DsPy). Therefore, the goal of our work is not to *scale up* test-time computation, as argued in prior work (Jin et al., 2025a; Chen et al., 2025a), but rather to *adaptively control* it based on the difficulty of each question.

Our framework, FRUGALRAG requires access only to ground truth documents $Y$ during training. It does **not** require supervision in the form of final answer annotations. These documents are used to provide fine-grained feedback signals. At inference time, FRUGALRAG relies solely on the input question $Q$, the document index $\mathcal{I}$, and a trained retriever $\mathcal{R}$.

In the following subsections, we describe the two stages of our learning algorithm: (1) **Stage 1** (Sec. 3.1): Generating a base policy that maximizes evidence coverage through exploration, and (2) **Stage 2** (Sec. 3.2): Finetuning this policy with reinforcement learning to control test-time compute.

## 3.1 STAGE 1: EVIDENCE COVERAGE MAXIMIZATION (EXPLORE)

Gathering evidence plays a crucial role in answering multi-hop questions, which often require iterative retrieval and reasoning across multiple sources. Drawing on recent advances in test-time scaling, we observe that we can boost evidence coverage (i.e., recall) simply by letting the model $f$ issue multiple search queries $S_h$ at inference time. This approach sidesteps the need for massive supervised fine-tuning—instead, it harnesses the model's own generated rollouts to gather, and then integrate additional information. In the next section, we describe how we construct our training data and design our fine-tuning protocol to fully leverage this capability.

**Training Dataset Generation.** We refer to a rollout as a set of outputs generated by the model $f$. Each rollout comprises a thought-action pair $(T_h A_h)$ at each hop $h \in [1, B]$, where $A_h$ is either a call to the retriever $\mathcal{R}$ or FINISH indicating the end of rollout. This setup is similar to the standard ReAct (Yao et al., 2023) framework which provides a general prompting template for interleaving external (retrieved) information and model generated outputs. At each hop $h$, we generate samples $\{(T_h^1, A_h^1, S_h^1) \ldots (T_h^n, A_h^n, S_h^n)\}$ using $n$ bootstrapped prompts (Khattab et al., 2023) (See Appendix F). For each search query $S_h^i, i \in [1, n]$ we retrieve corresponding documents $\mathcal{D}_h^i = \mathcal{R}(S_h^i)$, then discard any documents already present in the context. We then compute recall against ground-truth labels and add the sample $i$ that achieves the highest recall to the context for the next hop $h + 1$. This dataset generation strategy is simple and easily parallelizable. We conduct two separate runs – a standard run where $f$ is allowed terminate generation by generating FINISH, and the second where $f$ can only call the retriever till maximum budget is reached. Although the former is more efficient and finishes before B search hops, we observe that the latter yields a higher overall recall owing to a greater number of retrievals. Unlike previous work (Chan et al., 2024; Asai et al., 2023; Hsu et al., 2024; Jin et al., 2025a) that generated orders of magnitude more data, we only generate 1000 examples during this step.

**Supervised "Exploration" Finetuning (FRUGALRAG-Explore).** Although the base model $f$ without FINISH maximizes exploration, we cannot use it directly for reinforcement learning because it does not include FINISH. Consequently, during fine-tuning, we sample rollouts from both the configurations described above, i.e., 90% without FINISH and 10% with it. We want to use supervised finetuning to build a strong base-policy for RL, that prioritizes exploration while ensuring that FINISH remains in the model's generation distribution. Hence, we finetune the model $f$ to obtain our base policy $f_S$. At each iteration, the model predicts the next $(T_h, A_h, S_h)$ tuple given the rollout which comprises interleaved thought-action-search tuples and retrieved documents represented as an ordered set $\{(\mathcal{D}_0, T_0, A_0, S_0), (\mathcal{D}_1, T_1, A_1, S_1) \ldots (\mathcal{D}_{(h-1)}, T_{(h-1)}, A_{(h-1)}, S_{(h-1)})\}$ till $h-1$, using standard cross-entropy error as the objective function. $f_S$ has several notable advantages – **(1)** *off-the-shelf model $f$ does not explore.* Despite prompt optimization, $f$ is generally over-confident and predicts the answer without sufficient exploration. **(2)** *removing* FINISH *results in over-retrievals.* We observe that simply removing FINISH from the action space yields high recall even with the base model $f$, however, the model is forced to utilize the full budget for every question and cannot be post-trained for efficiency as it never generates a rollout with FINISH.

## 3.2 STAGE 2: CONTROLLING TEST-TIME COMPUTE WITH RL

Given a finetuned base policy model $f_S$, we propose a strategy that enables the model to generate extended rollouts *only when required*. This mechanism allows the model to adaptively determine the appropriate rollout length based on the complexity of the question, rather than scale the number of retrievals as in recent RL techniques (Jin et al., 2025a). Since $f_S$ generally prioritizes exploration, our only goal is to learn when to sufficient evidence has been gathered. In turn, we can also reduce the overall search latency during inference whenever possible. Below we show how this problem can be formulated as a reinforcement learning task, as it requires evaluating and comparing different rollouts.

**Reward Design.** Our reward function is designed to guide the model toward discovering the optimal rollout length. To compute the reward, we first generate the complete rollout using the policy $f_S$ and then evaluate it with ground truth evidence. Let $h^*$ denote the optimal number of retrieval steps (or hops), defined as the point beyond which further retrievals do not improve a predefined metric, say $c$. We compute $c$, the document recall against ground-truth evidence $Y$. Operationally, $h^*$ is the smallest hop at which the rollout achieves recall $\geq \tau$.

If the model terminates at a hop $h_{\text{term}} > h^*$, it incurs a penalty for redundant steps. Conversely, stopping at $h_{\text{term}} < h^*$ is also penalized to encourage sufficient exploration. This reward structure enables the model to explore adequately for complex queries while avoiding unnecessary computation for simpler ones. Mathematically,

$$
\mathbf{R} = \begin{cases} \max\left(-R_{\max}, \min\left(\log\left(\frac{1-\Delta}{\Delta}\right), R_{\max}\right)\right), & \text{if } \Delta > 0 \wedge c \geq \tau \quad \text{(late stop)} \\ R_{\max} + \alpha \cdot \left(\frac{h^*}{B}\right), & \text{if } \Delta = 0 \wedge c \geq \tau \quad \text{(perfect stop)} \\ \max\left(-R_{\max}, \min\left(\log\left(\frac{1-\Delta}{\Delta}\right), 0\right)\right), & \text{if } c < \tau \quad \text{(early stop)} \end{cases} \quad (1)
$$

Here, $\Delta$ denotes the absolute normalized deviation

$$
\Delta = \frac{|h_{\text{term}} - h^*|}{B},
$$

where $B$ is the maximum hop budget. When $c < \tau$, we set $h_{\text{term}} = B$ to promote continued exploration rather than early termination. Note that $\Delta$ is clipped for numerical stability.

$R_{\max}$ specifies the upper bound on the reward, and $\alpha$ is a tunable hyperparameter that scales the bonus awarded for terminating exactly at $h^*$. This bonus is proportional to $\frac{h^*}{B}$, thereby assigning greater reward to correct terminations on more complex (i.e., longer-horizon) rollouts. A rollout is deemed answerable if its performance satisfies $c \geq \tau$.

Table 1: Results on HotPotQA, 2Wiki, and MuSiQue with ColBERTv2 retriever. FRUGALRAG demonstrates superior performance on Model Based Evaluation (MBE) and Gold Evidence Recall (%) compared to standard baselines. Here, Qwen2.5-7B-Instruct is used as the answer generation model. Best results are in **bold** and second best are underlined.

| Method | HotPotQA | | | 2Wiki | | | MuSiQue | | |
|---|---|---|---|---|---|---|---|---|---|
| | MBE | Recall | Searches | MBE | Recall | Searches | MBE | Recall | Searches |
| Zero-Shot | 28.10 | NA | NA | 28.4 | NA | NA | 10.6 | NA | NA |
| Zero-Shot CoT | 29.10 | NA | NA | 29.8 | NA | NA | 10.7 | NA | NA |
| Zero-Shot RAG | 51.00 | 59.05 | 1 | 28.8 | 34.75 | 1 | 18.5 | 23.96 | 1 |
| ReAct + DsPy | 64.20 | 79.60 | 2.76 | 45.60 | 58.5 | 3.37 | 29.20 | 52.1 | 3.37 |
| **FRUGALRAG-Explore 7B** +*Qwen2.5-7B-Inst* | 67.70 | **86.40** | 5.99 | 47.60 | **68.90** | 5.99 | 30.10 | **59.10** | 5.93 |
| **FRUGALRAG 7B** +*Qwen2.5-7B-Inst* | **68.47** | 82.80 | 2.05 | **48.93** | 63.50 | 2.95 | **33.72** | 54.00 | 3.02 |

Intuitively, the reward function imposes a penalty proportional to $|\Delta|$, which quantifies the deviation from the optimal stopping step. Accordingly, the reward decreases monotonically as the termination step moves away from $h^*$, and it is maximized when $h_{\text{term}} = h^*$.

In addition to $\mathbf{R}$ (Eq. 1), we define a format reward $\mathbf{R_f}$ to enforce adherence to the format. If the output deviates from the expected format and results in failed retrievals, we assign a reward of $-0.5$; conversely, if the retrievals succeed, the model receives a reward of $+0.5$. This reward is averaged across all hops. The final reward is the mean of the main reward $\mathbf{R}$ and the format reward $\mathbf{R_f}$, yielding an overall reward in the range $[-R_{\max} - 0.50, \ R_{\max} + \alpha + 0.50]$.

**Optimal Rollout Length.** We define the optimal rollout length $h^*$ as the minimum number of retrieval steps required to answer a query $Q$ effectively. Any additional retrievals beyond $h^*$ are considered redundant. Conversely, if the rollout has not yet gathered sufficient evidence to answer the query, it should continue retrieving. To balance retrieval efficiency and performance, we use FRUGALRAG-Explore ($f_S$) as a reference policy, assuming it represents the best achievable performance within a fixed budget $B$. Similarly, we determine the threshold $\tau$ based on the performance of the base policy FRUGALRAG-Explore ($f_S$).

**Optimization.** We adopt GRPO (Shao et al., 2024) optimization algorithm because of its memory efficiency. At each hop $h$, we sample $v$ tuples $\{T_h^i, A_h^i, S_h^i\}_{i=1}^v$, and retrieve non-duplicate documents $\mathcal{D}_h^i$. We collect sample tuples and documents through this process until FINISH or till maximum budget is exhausted. For each rollout $i$, we then compute a cumulative reward using Eq. 1, and backpropagate through every logit produced by the policy along that rollout.

# 4 EXPERIMENTS

## 4.1 EXPERIMENTAL SETUP

**Benchmarks and Evaluation.** We evaluate FRUGALRAG on three widely adopted multi-hop retrieval-augmented generation (RAG) benchmarks—HotPotQA (Yang et al., 2018), 2WikiMultiHopQA (Ho et al., 2020) (2Wiki), and MuSiQue (Trivedi et al., 2022b)—under their full-wiki setting using the development splits, following the protocol of FlashRAG (Jin et al., 2025b). Beyond multi-hop RAG, we also assess FRUGALRAG in a highly challenging deep research scenario using BrowseCompPlus (Chen et al., 2025b). To demonstrate the effectiveness of FRUGALRAG, we experiment with three different retrievers: ColBERT-v2 (Santhanam et al., 2021), Qwen3-8B-Embedding (Zhang et al., 2025), and E5-base-v2 (Wang et al., 2022). Additional details on the datasets and indexing are provided in Appendix G. Following prior work, we adopt a robust LLM judge-based evaluation metric (Chen et al., 2025b) to assess the accuracy of the final answers (MBE - Model Based Evaluation). We choose Qwen3-32B for answer evaluation. For retrieval-level evaluation, we follow FlashRAG (Jin et al., 2025b) and use recall as the metric, which measures whether the ground truth answer is present in the retrieved documents, assigning a score of 1 if it is found.

Table 2: Comparison of FRUGALRAG against state-of-the-art approaches on HotPotQA, 2Wiki, and MuSiQue, evaluated using Answer (MBE) and Retrieval (Recall) metrics with the E5-base-v2 retriever. Despite being trained on only 1,000 examples, FRUGALRAG delivers consistently competitive performance. The results further show that FRUGALRAG is compatible with diverse answer generator models, demonstrating strong modularity. Best results are in **bold** and second best are underlined.

| Method | #Train | HotPotQA | | | 2Wiki | | | MuSiQue | | |
|---|---|---|---|---|---|---|---|---|---|---|
| | | MBE | Recall | Searches | MBE | Recall | Searches | MBE | Recall | Searches |
| Zero-Shot | – | 28.10 | NA | NA | 28.4 | NA | NA | 10.60 | NA | NA |
| Zero-Shot CoT | – | 29.10 | NA | NA | 29.8 | NA | NA | 10.7 | NA | NA |
| Zero-Shot RAG | – | 45.50 | 52.49 | 1 | 28.80 | 35.00 | 1 | 18.00 | 22.01 | 1 |
| ReAct + DsPy | 100 | 49.30 | 63.40 | 3.23 | 35.65 | 60.03 | 3.27 | 27.30 | 49.8 | 3.74 |
| IRCOT | – | 45.60 | 66.90 | 3.31 | 30.53 | 57.80 | 3.41 | 20.62 | 44.10 | 3.19 |
| ITER-RETGEN | 36k | 46.10 | 55.40 | 3.00 | 32.10 | 45.30 | 3.00 | 19.10 | 34.20 | 3.00 |
| Ret-Robust | 1000 | 39.38 | 35.90 | 4.02 | 45.17 | 31.50 | 4.23 | 24.24 | 18.80 | 4.23 |
| Self RAG 7B | 150k | 37.60 | 52.60 | 1.00 | 30.10 | 41.30 | 1.00 | 15.00 | 26.70 | 1.00 |
| Self RAG 13B | 150k | 39.70 | 52.60 | 1.00 | 31.50 | 41.30 | 1.00 | 15.00 | 26.70 | 1.00 |
| O2 Searcher | 1440 | 42.70 | 50.10 | 1.77 | 45.70 | 55.20 | 2.42 | 26.40 | 37.00 | 1.95 |
| SimpleDeepSearcher | 871 | 50.40 | 64.80 | 2.75 | 49.30 | 60.50 | 3.64 | 34.30 | 50.40 | 2.86 |
| R1-Searcher | 8k | 57.66 | 69.10 | 2.22 | 52.00 | 60.40 | 2.36 | 39.78 | **57.70** | 2.31 |
| Search-R1 | >100k | 46.20 | 48.20 | 1.28 | 36.20 | 47.70 | 1.89 | 24.80 | 38.10 | 1.36 |
| CoRAG | >100k | 58.20 | 64.30 | 4.00 | **59.00** | **65.40** | 4.00 | **40.50** | 54.00 | 4.00 |
| **FRUGALRAG-7B** +*Qwen2.5-7B-Inst* | 1000 | 58.5 | **70.40** | 2.89 | 50.40 | 58.80 | 3.03 | 36.40 | 53.30 | 3.30 |
| **FRUGALRAG-7B** + *Qwen2.5-32B-Inst* | 1000 | **61.4** | **70.40** | 2.89 | 50.90 | 58.80 | 3.03 | 39.90 | 53.30 | 3.30 |
| **FRUGALRAG-7B** + *CoRAG* | 1000 | 58.00 | **70.40** | 2.89 | 51.20 | 58.80 | 3.03 | 34.60 | 53.30 | 3.30 |

Finally, to evaluate efficiency, we report the number of searches performed as a proxy for latency across different methods.

**Baselines.** We perform extensive experiments comparing FRUGALRAG with several strong baselines and state-of-the-art models. Our evaluation covers: (1) zero-shot performance; (2) prompting-based methods such as ReAct (Yao et al., 2023), IRCOT (Trivedi et al., 2022a), and DsPy (Khattab et al., 2023); (3) finetuned approaches including Iter-RetGen (Shao et al., 2023), Ret-Robust (Yoran et al., 2024), and Self-RAG (Asai et al., 2023). We further benchmark FRUGALRAG against recent reasoning-focused models, including CoRAG (Wang et al., 2025), O2 Searcher (Mei et al., 2025), Simple Deep Searcher (Sun et al., 2025), and R1-Searcher (Song et al., 2025).

**Training and Inference.** FRUGALRAG is model-agnostic; we train Qwen2.5-7B-Instruct using our two-stage framework. Both supervised finetuning and reinforcement learning (RL) stages leverage the TRL library (von Werra et al., 2020), with prompt bootstrapping based on DsPy (Khattab et al., 2023). For each dataset, Stage 1 finetuning uses 1000 randomly sampled training examples, performing full-parameter training for one epoch with a learning rate of $2 \times 10^{-5}$, weight decay of 0.01, and a maximum sequence length of 4096 (Sec. 3.1). Stage 2 applies GRPO (Shao et al., 2024) to further train the models (Sec. 3.2). During inference, we use our finetuned model FRUGALRAG for multi-hop reasoning and retrieval followed by an off-the-shelf model for final answer generation. The answer generator is provided with the necessary context (trajectory-level information) to generate the final answer. Full hyperparameter and prompt details are in Appendices F, H respectively.

## 4.2 MAIN RESULTS: EFFECTIVENESS OF FRUGALRAG

Table 1 compares FRUGALRAG with standard zero-retrieval and prompt-based baseline (ReAct). For fair comparison, all baselines use the ColBERTv2 (Santhanam et al., 2021) retriever indexed on Wikipedia (See Appendix F) and Qwen2.5-7B-Instruct as the base model. The key takeaway is that FRUGALRAG consistently outperforms the baselines on both answer and retrieval metrics, using competitive or significantly smaller number of searches on average.

Table 3: BrowseCompPlus. FRUGALRAG, trained on HotPotQA generalizes to this harder task and outperforms significantly larger models such as DeepSeek-R1.

| Method | Model Size | Accuracy (%) | Recall (%) | Avg. Searches |
|---|---|---|---|---|
| Sonnet 4 | - | 37.35 | 47.33 | 9.03 |
| Opus 4 | - | 36.75 | 50.84 | 10.24 |
| kimi-k2-0711-preview | - | 35.42 | 38.38 | 11.22 |
| Gemini-2.5-Flash | - | 34.58 | 40.19 | 9.77 |
| Gemini-2.5-Pro | - | 29.52 | 35.31 | 6.04 |
| oss-120b-low | 120B | 25.54 | 22.50 | 2.21 |
| oss-20b-high | 20B | 35.06 | 49.29 | 23.87 |
| oss-20b-medium | 20B | 30.48 | 41.31 | 13.64 |
| oss-20b-low | 20B | 14.10 | 17.37 | 1.87 |
| DeepSeek-R1-0528 | 600B | 16.39 | 16.32 | 2.72 |
| Qwen3-32B | 32B | 10.72 | 7.80 | 0.94 |
| Search-R1-32B | 32B | 11.08 | 10.17 | 1.69 |
| **FRUGALRAG-7B +** *Qwen2.5-7B-Inst* (HotPotQA) | 7B | 20.46 | 23.57 | 7.95 |
| **FRUGALRAG-7B +** *Qwen2.5-7B-Inst* (2Wiki) | 7B | 21.53 | 22.93 | 10.96 |
| **FRUGALRAG-7B +** *Qwen2.5-7B-Inst* (MuSiQue) | 7B | 21.14 | 23.73 | 8.39 |

The *ReAct Few Shot (FS) baseline* (Yao et al., 2023; Khattab et al., 2023) outperforms vanilla Retrieval Augmented Generation. The optimized prompts enable ReAct to generate more search queries, and as a result achieve very strong performance. Strikingly, on HotPotQA, ReAct achieves (a) **79.60%**, which is greater than the recall achieved by LeReT (Hsu et al., 2024) (77.1% using Llama3.1-8B) after significant finetuning; signifying the importance of building a strong baseline.

Table 1 demonstrates the overall *effectiveness of* FRUGALRAG-*Explore* achieving the highest Recall and MBE scores compared to all baselines. However, we note that FRUGALRAG-Explore is either best or second best in terms of both answer and recall metrics but introduces a very high latency compared to FRUGALRAG. Stage-2 finetuning significantly reduces the number of searches required while retaining the performance across three datasets. For instance, the search count reduces on average by ≈50% across the three datasets while the MBE scores improve. Overall, our results highlight that FRUGALRAG strikes a significantly better efficiency-accuracy tradeoff.

## 4.3 COMPREHENSIVE BASELINE COMPARISONS

Table 2 demonstrates the performance of FRUGALRAG against 14 baselines and state-of-the-art RAG methods. For fair comparison, we setup our experiment using FlashRAG (Jin et al., 2025b) with a common retriever backend, E5-Base-V2 and an index of 21 million wikipedia passages.

FRUGALRAG achieves strong overall performance across all three benchmarks. On HotPotQA, it attains 58.5% MBE with Qwen2.5-7B and improves to 61.4% with Qwen2.5-32B while being trained on only 1000 examples. Notably, it also achieves the highest retrieval recall (70.4%), indicating that its gains are not merely due to stronger generation but are supported by improved evidence acquisition. In contrast, the second-best method, CoRAG, relies on a fixed inference-time budget, which may be inefficient for simpler questions and inflexible for more complex ones. FRUGALRAG instead adaptively allocates search steps, using approximately 2.89 searches on average (HotPotQA).

On 2Wiki and MuSiQue, FRUGALRAG remains competitive, ranking best or second-best in MBE while maintaining strong recall. Although CoRAG achieves the top MBE on these datasets, it relies on over 100k training examples, joint training of the answer generator, and a fixed search budget. In comparison, FRUGALRAG matches or closely approaches these results with two orders of magnitude fewer training examples and consistently strong retrieval performance. Additionally, FRUGALRAG uses an off-the-shelf answer generator without additional fine-tuning. This is important because for other methods, performance gains may partially stem from generator training rather than improvements in multi-hop retrieval. By decoupling retrieval from generation and still achieving

Table 4: FRUGALRAG issues more queries for harder questions. We see a clear increasing trend in num. of search queries with amount of ground truth evidence (2Wiki) and ground truth num. of hops.

| 2WikiMultiHopQA | | | MuSiQue | | |
|---|---|---|---|---|---|
| Num GT Evidence | Number of Questions | Searches | Actual Hops | Number of Questions | Searches |
| 2 | 9,595 | $2.665 \pm 1.430$ | 2 | 1,252 | $3.054 \pm 1.433$ |
| 3 | 88 | $2.875 \pm 1.483$ | 3 | 760 | $3.924 \pm 1.464$ |
| 4 | 2,806 | $3.909 \pm 1.502$ | 4 | 405 | $4.205 \pm 1.368$ |

Table 5: Comparison of SFT baselines and FRUGALRAG across datasets using the Efficiency Tradeoff metric (Eff) = $(\mathrm{MBE} + \mathrm{Recall})/(2 \times \mathrm{Searches})$. R: Recall, MBE: Model-Based Evaluation, S: Searches.

| Method | HotpotQA | | | | 2Wiki | | | | Musique | | | |
|---|---|---|---|---|---|---|---|---|---|---|---|---|
| | R | MBE | S | Eff. | R | MBE | S | Eff. | R | MBE | S | Eff. |
| *E5* | | | | | | | | | | | | |
| SFT (with FINISH) | 68.9 | 57.8 | 2.84 | **22.31** | 52.6 | 43.8 | 3.48 | 13.85 | 52.5 | 31.8 | 3.43 | 12.29 |
| FRUGALRAG-Explore-7B + *Qwen2.5-7B-Inst* | 77.0 | 59.60 | 10.88 | 6.28 | 64.3 | 46.80 | 10.79 | 5.15 | 51.7 | 35.30 | 10.85 | 4.01 |
| **FRUGALRAG-7B** + *Qwen2.5-7B-Inst* | 70.4 | 58.5 | 2.89 | 22.30 | 58.8 | 50.4 | 3.03 | **18.02** | 53.30 | 36.40 | 3.30 | **13.59** |
| *ColBERTv2* | | | | | | | | | | | | |
| SFT (with FINISH) | 79.90 | 67.70 | 2.07 | 35.65 | 63.70 | 48.50 | 3.13 | 17.93 | 51.70 | 29.80 | 3.25 | 12.54 |
| FRUGALRAG-Explore-7B + *Qwen2.5-7B-Inst* | 86.40 | 67.70 | 5.99 | 12.87 | 68.90 | 47.60 | 5.99 | 9.73 | 59.10 | 30.10 | 5.93 | 7.52 |
| **FRUGALRAG-7B** + *Qwen2.5-7B-Inst* | 82.80 | 68.50 | 2.05 | **36.90** | 63.50 | 48.90 | 2.95 | **19.05** | 54.00 | 33.70 | 3.02 | **14.52** |

competitive results, FRUGALRAG isolates the contribution of adaptive search and demonstrates that strong multi-hop reasoning can be obtained without end-to-end retraining of the generator. Finally, the improvements obtained by swapping the answer generator (e.g., from Qwen2.5-7B to Qwen2.5-32B or CoRAG) demonstrate the modularity of FRUGALRAG, confirming that its benefits stem from the search-and-retrieval strategy rather than reliance on a specific generator.

## 4.4 RESULTS ON DEEP RESEARCH

We evaluate FRUGALRAG on BrowseComp-Plus, a recent, challenging deep research dataset that demands massively multi-hop search and reasoning. Strikingly, our models trained HotPotQA, MuSiQue, and 2Wiki readily generalize to harder datasets like BrowseComp-Plus. In Table 3 we show that FRUGALRAG issues between 7-10 queries on average despite being trained with a budget of 6 queries. We show that FRUGALRAG achieves superior performance compared to many larger models like Search-R1 32B, Qwen3-32B (Yang et al., 2025), and DeepSeek-R1 (DeepSeek-AI, 2025) on both answer accuracy (MBE) and recall.

## 4.5 ANALYSIS

**FRUGALRAG is Adaptive.** Table 4 demonstrates that FRUGALRAG adaptively scales test-time compute—measured by the number of search queries—in proportion to question *difficulty*. To operationalize difficulty, we leverage MuSiQue, which provides explicit annotations for the expected number of reasoning hops, and 2Wiki, where we use the number of ground-truth evidence documents as a proxy for reasoning complexity. To rigorously assess the correlation between question hardness and the number of queries issued by FRUGALRAG, we evaluate over the *entire* development sets of both datasets (approx. 12k and 2.5k examples, respectively). We observe strong positive correlations: $r = 0.82$ on 2Wiki and $r = 0.95$ on MuSiQue. These results indicate that FRUGALRAG effectively allocates additional computation in response to increased multi-hop reasoning demands.

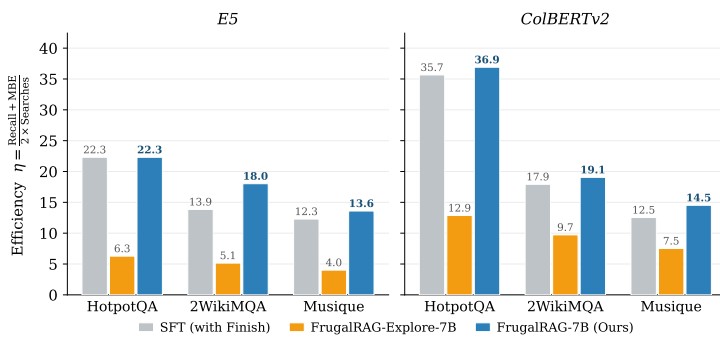

Figure 1: FRUGALRAG on average outperforms fixed budget and SFT based baselines, demonstrating the effectiveness of both Stage-1 finetuning and Stage-2 learning to control test time compute. The plots show the Tradeoff metrics (See Table 5 for detailed results).

**FRUGALRAG is robust.** In Table 14, we report results for models trained on 1000 examples from HotPotQA, 2Wiki, and MuSiQue and evaluated across all datasets, testing FRUGALRAG's ability to generalize to unseen scenarios. Despite the distribution shift, FRUGALRAG maintains strong performance without compromising efficiency. This robustness holds for both E5-base-v2 and ColBERT-v2 retrievers and different answer generators.

**FRUGALRAG outperforms SFT.** To capture the trade-off between answer quality and retrieval cost, we define a simple metric: Efficiency Tradeoff: $(MBE + Recall)/(2 * Searches)$. This metric allows us to jointly evaluate answer accuracy and evidence quality relative to the number of retrieval steps, enabling a comparison of efficiency across models and budgets. To assess the impact of the RL-finetuning (Stage 2), we conduct an ablation study by comparing FRUGALRAG against a simple baseline, called SFT (with FINISH). This baseline is trained for 1 epoch on the dataset generated during Stage 1, but with one key modification. Instead of sampling 90% rollouts without FINISH, we sample all traces where the model outputs FINISH on its own. In Fig. 1, we demonstrate that FRUGALRAG *on average across all models* is better on both Efficiency Tradeoff. FRUGALRAG outperforms both FRUGALRAG-SFT and FRUGALRAG-Explore by a significant margin using Qwen2.5-7B-Instruct on all the datasets as illustrated in Fig. 1.

## 5 CONCLUSIONS, LIMITATIONS, AND FUTURE WORK

In this work, we showed that reinforcement learning can be used to optimize the search queries issued by a RAG system, based on query difficulty. We found that a simple ReAct baseline that iteratively retrieves (by invoking the search tool) and reasons (to decide what search call to issue next) is quite competitive, especially if we can optimize its few-shot prompt using just tens of training examples. We proposed a two-stage framework FRUGALRAG that a) works with 1000 training examples, compared to state-of-the-art RAG techniques that use over 100,000 examples, and b) yet achieves competitive accuracies while also using far fewer search queries at inference time, on popular multi-hop QA datasets.

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

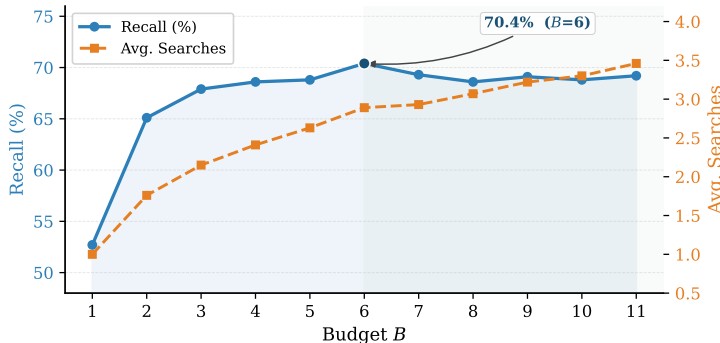

Figure 2: FRUGALRAG tested using different maximum budgets $B$ on HotPotQA. We find that our approach achieves maximum recall at the training budget followed by diminishing returns on subsequent searches.

## A ADDITIONAL RESULTS

### A.1 ADDITIONAL ABLATIONS

**Effect of changing Budget $B$.** We analyze the sensitivity of FRUGALRAG to the retrieval budget at inference time on HotPotQA. Specifically, we train with a maximum budget of $B = 6$ and evaluate it under a range of budgets $B \in \{1, \ldots, 11\}$. Figure 2 reports recall and the average number of searches executed.

As the budget increases, the model adaptively utilizes the additional compute by issuing more searches. Recall improves sharply when moving from very small budgets ($B = 1$–$2$) to moderate budgets ($B = 3$–$6$), indicating that additional retrieval steps in this regime meaningfully expand the available evidence. Beyond $B = 6$, improvements become marginal despite a steady increase in search count. This saturation behavior is consistent with diminishing returns from additional retrieval once sufficient supporting evidence has been gathered.

Notably, the highest recall is achieved at $B = 6$, which matches the training-time budget. This is expected: the policy has learned retrieval strategies calibrated to this constraint. Similar saturation trends are observed on 2WikiMultiHopQA and MuSiQue. On substantially more complex datasets, however, larger budgets may continue to yield gains. Exploring training under variable or adaptive budgets is a promising direction for future work.

**Sample Count Ablation.** We analyze the impact of the number of supervised training examples on retrieval performance and efficiency in Fig. 3. Across all three datasets, recall improves substantially as the number of training examples increases from 100 to 500. For instance, on HotPotQA, recall improves from 67.9 to 70.8 when scaling from 100 to 500 examples. The same upward trend is observed on 2WikiMultiHopQA and MuSiQue, indicating that the policy rapidly acquires the core retrieval and termination behaviors with only a few hundred annotated instances.

Beyond 500 examples, gains become more incremental but remain consistent. Notably, search efficiency stabilizes—and in some cases improves—with additional data. At 5,000 training examples, the average number of searches decreases consistently across all three datasets, suggesting that the policy not only improves recall but also learns to allocate retrieval steps more judiciously.

Overall, these results indicate that FRUGALRAG operates effectively in the low-data regime, achieving strong performance with relatively limited supervision while continuing to benefit from additional examples. Nevertheless, these findings should be interpreted with care, our experiments focus primarily on small-scale supervision, and a comprehensive study of large-scale scalability is a future direction.

**FRUGALRAG is agnostic to base model.** To evaluate whether FRUGALRAG generalizes beyond the Qwen2.5-7B-Instruct base model, we conduct additional experiments using Llama3.1-8B-Instruct.

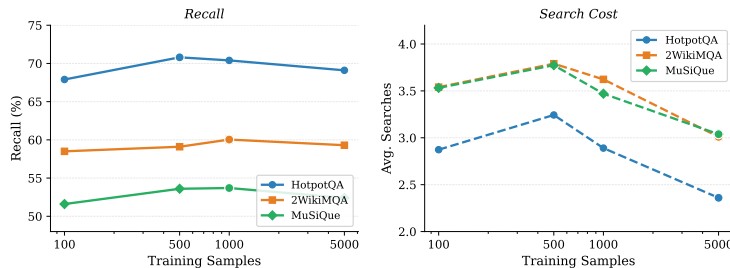

Figure 3: FRUGALRAG trained with varying number of supervised examples. We demonstrate that FRUGALRAG is robust even in low-data regimes and performance improves consistently with increasing number of examples.

| Base Model | Dataset | Recall (%) | Avg. Searches |
|---|---|---|---|
| FrugalRAG (Qwen2.5-7B-Instruct) | HotPotQA | 70.40 | 2.89 |
| FrugalRAG (Llama3.1-8B-Instruct) | HotPotQA | 70.00 | 2.95 |
| FrugalRAG (Qwen2.5-7B-Instruct) | 2WikiMultiHopQA | 60.40 | 3.62 |
| FrugalRAG (Llama3.1-8B-Instruct) | 2WikiMultiHopQA | 60.41 | 3.79 |
| FrugalRAG (Qwen2.5-7B-Instruct) | MuSiQue | 53.70 | 3.74 |
| FrugalRAG (Llama3.1-8B-Instruct) | MuSiQue | 53.00 | 3.96 |

Table 6: Performance comparison across different base models.

In this setting, we reuse the Stage-1 supervised training data originally generated with Qwen2.5-7B-Instruct and fine-tune Llama3.1-8B-Instruct under the same two-stage pipeline.

As shown in Table 6, performance remains highly consistent across base models on all three datasets. Recall differences are marginal (within ∼0.7 points), and search behavior remains comparable. These results indicate that FRUGALRAG is model-agnostic and that the gains stem from the training framework.

## B    REWARD DESIGN

We establish the efficacy of FRUGALRAG by conducting additional reward design ablations.

**Stage-1 is necessary.** We first evaluate *Stage-2 without Stage-1* on HotPotQA, where we directly apply RL fine-tuning to the base Qwen2.5-7B-Instruct policy without prior SFT. As shown in Table 7, removing Stage-1 substantially degrades recall (66.5 vs. 70.4). This demonstrates that SFT is a crucial step that teaches the model essential reasoning and exploration, stabilizing the subsequent RL optimization which uses the proposed "learning to stop" criterion.

**Alternative rewards.** We analyze alternative reward formulations and training strategies on HotPotQA. Table 8 summarizes the results. Directly training Qwen2.5-7B-Instruct with a recall reward yields substantially worse performance (66.8%), showing that naively applying RL does not recover the behavior of FRUGALRAG. Moreover, optimizing recall alone in FRUGALRAG-EXPLORE dramatically increases search usage (5.76), indicating poor cost control. Furthermore, jointly optimizing both query generation and stopping via recall reward, overall recall decreases (68.3%), suggesting that simultaneous optimization of both behaviors increases training difficulty and leads to suboptimal convergence. Finally, Table 8 demonstrates that removing the late-stop penalty yields similar recall (70.5%) but increases the average number of searches (3.144 vs. 2.89). This verifies that the penalty term effectively removes redundant retrieval steps without sacrificing accuracy. Across all ablations, FRUGALRAG consistently achieves the strongest balance between recall and search efficiency. These experiments demonstrate that the improvements are not merely the result of generic RL regularization, but stem from (i) the two-stage training procedure and (ii) the specific reward design that explicitly balances recall gains against retrieval cost.

| Ablation | Recall (%) | Avg. Searches |
|---|---|---|
| Stage-2 w/o Stage-1 (w/o SFT) | 66.5 | 2.36 |
| **FRUGALRAG (Ours)** | **70.4** | 2.89 |

Table 7: Effect of removing Stage-1 supervised fine-tuning.

| Ablation | Recall (%) | Avg. Searches |
|---|---|---|
| Qwen2.5-7B-Instruct + Recall Reward | 66.8 | 2.57 |
| FrugalRAG-Explore + Recall Reward | 76.0 | 5.76 |
| FrugalRAG + Recall Reward (Query Opt) | 68.3 | 2.45 |
| FrugalRAG w/o Late Penalty | 70.5 | 3.144 |
| **FrugalRAG (Ours)** | **70.4** | 2.89 |

Table 8: Reward and optimization ablations on HotPotQA.

## B.1 COMPARISON AGAINST HEURISTIC STOPPING RULES

To further isolate the contribution of our learned termination policy, we compare FRUGALRAG against fixed heuristic stopping strategies that do not require training a separate model. This ensures that any performance differences are attributable to our reward design rather than regularization. Other heuristic-style stopping mechanisms (e.g., ReAct (Yao et al., 2023) and IRCOT (Trivedi et al., 2022a)) are already included as baselines in Table 2 of the main paper. Here, we focus specifically on fixed-depth stopping rules derived from dataset statistics. Table 9 summarizes the results.

1. **Stopping Heuristic 1 (Peak-Recall Depth).** We first estimate the retrieval depth at which recall peaks on the training set under the FRUGALRAG-EXPLORE policy. The estimated depths are: HotPotQA (2.96), 2WikiMultiHopQA (1.93), and MuSiQue (4.96). These values are rounded to the nearest integer and used as fixed stopping depths at test time.

2. **Stopping Heuristic 2 (Ground-Truth Average Depth).** In a second variant, we compute the average number of retrieval steps required to recover the ground-truth evidence chains in the training set: HotPotQA (2.0), 2WikiMultiHopQA (2.40), and MuSiQue (2.32). These averages are rounded to the nearest integer and used as constant stopping depths during inference.

Across HotPotQA, 2WikiMultiHopQA, and MuSiQue, fixed-depth heuristics consistently underperform the learned FRUGALRAG policy. Although heuristic stopping based on peak-recall depth can approach competitive performance on easier datasets (e.g., HotPotQA), it fails to match the efficiency–accuracy trade-off achieved by FRUGALRAG. In particular, on more challenging datasets such as 2WikiMultiHopQA and MuSiQue, where the optimal number of reasoning hops varies substantially across instances, the gap widens.

These results demonstrate that a static stopping rule, even when calibrated using training statistics or ground-truth evidence chains, cannot capture instance-specific termination dynamics. In contrast, our RL-based policy adaptively determines when to stop retrieval conditioned on the evolving reasoning state, yielding consistently higher recall with competitive or lower search budgets.

## B.2 ADDITIONAL EFFICIENCY ANALYSIS

We additionally compare computational efficiency across methods by reporting total processed tokens, approximate FLOPs, end-to-end latency, and average search count. Tables 10, 11, and 12 summarize the results on HotPotQA, 2WikiMultiHopQA, and MuSiQue, respectively.

Across datasets, FRUGALRAG remains competitive on compute-normalized metrics, including total tokens, FLOPs, and search count. While some baselines achieve lower latency due to fewer overall decoding steps, they typically do so at the cost of substantially lower recall. In contrast, FRUGALRAG consistently operates in a favorable efficiency–accuracy regime, matching or exceeding the best-performing baselines in recall while maintaining comparable token usage and search budgets.

| Dataset | Method | Recall | Avg. Searches |
|---------|--------|--------|---------------|
| HotPotQA | Heuristic 1 (Qwen2.5-7B-Instruct) | 67.7 | 3 |
| | Heuristic 1 (FrugalRAG-Explore) | 70.9 | 3 |
| | Heuristic 2 (FrugalRAG-Explore) | 67.9 | 2 |
| | **FrugalRAG (Ours)** | **70.4** | 2.89 |
| 2WikiMultiHopQA | Heuristic 1/2 (Qwen2.5-7B-Instruct) | 52.0 | 2 |
| | Heuristic 1/2 (FrugalRAG-Explore) | 55.4 | 2 |
| | **FrugalRAG (Ours)** | **58.8** | 3.03 |
| MuSiQue | Heuristic 1 (Qwen2.5-7B-Instruct) | 51.8 | 5 |
| | Heuristic 1 (FrugalRAG-Explore) | 54.5 | 5 |
| | Heuristic 2 (FrugalRAG-Explore) | 43.8 | 2 |
| | **FrugalRAG (Ours)** | **53.3** | 3.30 |

Table 9: Comparison between FRUGALRAG and fixed heuristic stopping strategies.

| HotPotQA | | | | | |
|----------|----------|-------|-------------|--------------|------------|
| Method | Avg Tokens | FLOPs | Latency (s) | Search Count | Recall (%) |
| O2 Searcher-3B | 2,937 | $1.79 \times 10^{13}$ | 0.1050 | 1.77 | 50.10 |
| SimpleDeepSearcher-7B | 9,657 | $1.31 \times 10^{14}$ | 0.6571 | 2.75 | 64.80 |
| R1 Searcher-7B | 4,458 | $6.17 \times 10^{13}$ | 0.1335 | 2.22 | 69.10 |
| Search-R1-7B | 2,119 | $2.99 \times 10^{13}$ | 0.0650 | 1.28 | 48.20 |
| CoRAG-8B | 7,770 | $1.24 \times 10^{14}$ | 0.1486 | 4.00 | 64.30 |
| **FrugalRAG-7B (ours)** | 9,138 | $1.27 \times 10^{14}$ | 0.2415 | 2.89 | 70.40 |

Table 10: Efficiency comparison on HotPotQA.

Importantly, in practical settings, monetary cost is dominated primarily by (i) the number of tokens generated and (ii) the number of external tool calls (i.e., search operations). Under these cost-relevant metrics, FRUGALRAG is directly comparable to—and in several cases more efficient than—strong multi-hop retrieval baselines. This indicates that the gains in retrieval effectiveness do not come from disproportionate increases in computational or tool usage, but rather from more effective, adaptive allocation of retrieval steps.

## C  ADDITIONAL EVALUATIONS

In multi-hop QA, retrieval differs from standard IR because of sequential queries rather than producing a single ranked list. The key metric in multi-hop retrieval is recall over all the steps, since missing any required evidence makes the reasoning worse. Prior work like CoRAG and Search-R1 only focus on answer-level metrics. To strengthen our evaluation, we report document-level overall precision and F1 in addition to recall. These metrics capture retrieval quality without relying on ranking assumptions that do not hold in the multi-hop setting. Table 13 summarizes our results.

## D  STABILITY OF RL TRAINING

We conduct three independent Stage 2 runs using 1000 examples from the HotPotQA datasets and test on all three dataset. The recall on HotPot evaluation is $69.60 \pm 1.21$ using $2.74 \pm 0.16$ searches on average. On 2WikiMultiHopQA and MuSiQue we achieve a recall of $59.20 \pm 1.11$ (searches - $3.60 \pm 0.11$) and $53.10 \pm 1.4$ (searches - $3.42 \pm 0.08$) respectively.

## E  QUALITATIVE EXAMPLES

In Table 4, we show that with increasing difficulty, FRUGALRAG on average adaptively issues more queries. To further this claim, we provide qualitative examples on MuSiQue in Table 15.

**2WikiMultiHopQA**

| Method | Avg Tokens | FLOPs | Latency (s) | Search Count | Recall (%) |
|---|---|---|---|---|---|
| O2 Searcher-3B | 4,212 | $2.53 \times 10^{13}$ | 0.1206 | 2.42 | 55.20 |
| SimpleDeepSearcher-7B | 13,433 | $1.80 \times 10^{14}$ | 0.7625 | 3.64 | 60.50 |
| R1 Searcher-7B | 4,939 | $6.89 \times 10^{13}$ | 0.1374 | 2.36 | 60.40 |
| Search-R1-7B | 3,456 | $4.96 \times 10^{13}$ | 0.0852 | 1.89 | 47.70 |
| CoRAG-8B | 8,018 | $1.28 \times 10^{14}$ | 0.1511 | 4.00 | 65.40 |
| **FrugalRAG-7B (ours)** | 11,279 | $1.45 \times 10^{14}$ | 0.4673 | 3.03 | 59.90 |

Table 11: Efficiency comparison on 2WikiMultiHopQA.

**MuSiQue**

| Method | Avg Tokens | FLOPs | Latency (s) | Search Count | Recall (%) |
|---|---|---|---|---|---|
| O2 Searcher-3B | 2,923 | $1.74 \times 10^{13}$ | 0.1009 | 1.95 | 37.00 |
| SimpleDeepSearcher-7B | 10,027 | $1.35 \times 10^{14}$ | 0.7366 | 2.86 | 50.40 |
| R1 Searcher-7B | 4,665 | $6.47 \times 10^{13}$ | 0.1327 | 2.31 | 57.70 |
| Search-R1-7B | 2,212 | $3.14 \times 10^{13}$ | 0.0676 | 1.36 | 38.10 |
| CoRAG-8B | 7,683 | $1.23 \times 10^{14}$ | 0.1472 | 4.00 | 54.00 |
| **FrugalRAG-7B (ours)** | 11,914 | $1.66 \times 10^{14}$ | 0.4098 | 3.30 | 52.60 |

Table 12: Efficiency comparison on MuSiQue.

# F  TRAINING DETAILS

Algorithm 1 shows the overall training framework of FRUGALRAG. We plan to publicly release our code soon. Below, we discuss each step along with their implementation details.

**Few-Shot Prompt Optimization Details.** We leverage DSPy (Khattab et al., 2023) for automatic few-shot prompt generation following LeReT (Hsu et al., 2024). Specifically, we use 100 training examples ($\mathcal{L}_{\text{init}}$) with the BOOTSTRAPFEWSHOTWITHRANDOMSEARCH method, which uses the LM $f$ to generate few-shot examples, selecting the best performing ones for subsequent prompting. Using more advanced prompting techniques may yield further improvements. We select 4 best performing few-shot prompts from a total of 15 candidate sets using the sum of answer EM and answer passage match. Answer EM checks for an exact string-match between the generated and actual answer, and passage match checks if the actual answer is present in the retrieved passages. This step is crucial because it facilitates dataset generation using diverse rollouts and ensures the answer format is followed by the model. For this step, we serve our model on one GPU using VLLM (Kwon et al., 2023).

**Dataset Generation Details.** For each few-shot prompt $p_i$, the model $f$ generates a tuple $(T_h^i, A_h^i, S_h^i)$ representing a candidate output for the next hop. As described in Sec. 3.1, we evaluate all candidate tuples at hop $h$ and select one with the highest recall. This selected candidate is then used as the context for the next hop and the process is repeated till budget $B$ (optionally till the selected candidate action $A_h$ indicates FINISH). We set the budget $B = 6$, where the initial retrieval step is always $\mathcal{R}(Q^{(j)})$ with $Q^{(j)}$ denoting the original user utterance. The generated dataset is denoted by $\mathbf{D}$. For all experiments involving Qwen2.5, we utilize the 7B-Instruct variant along with its prompts to generate the dataset. For further improving results, we can repeat few shot prompt optimization and dataset generation using different base models.

**Supervised "Explore" Finetuning Details.** We use the standard next token prediction loss given by:

$$\max_f \mathbb{E}_{(x,y) \sim \mathbf{D}} \log p_f(x|y) \tag{2}$$

where $y = (T_h, A_h, S_h)$ and $x = Q^{(j)} \cup \{T_k, A_k, S_k, \mathcal{D}_k\}_{k=0}^{h-1}$ sampled from the generated dataset $\mathbf{D}$.

We train the model $f$ for 1 epoch using a batch size of 4 and apply gradient accumulation of 2 steps, resulting in an effective batch size of 8. Optimization is performed using AdamW (Loshchilov and

| Dataset | Model | Rec | Prec | F1 | Searches |
|---------|-------|-----|------|-----|----------|
| HotPotQA | O2 Searcher | 50.10 | 22.29 | 30.85 | 1.77 |
| | SimpleDeepSearcher | 64.80 | 23.43 | 34.42 | 2.75 |
| | R1 Searcher | 69.10 | 24.25 | 35.90 | 2.22 |
| | Search-R1 | 48.20 | 20.24 | 28.51 | 1.28 |
| | CoRAG | 64.30 | 24.03 | 34.99 | 4.00 |
| | **FrugalRAG (ours)** | 70.40 | 26.81 | 38.83 | 2.89 |
| 2WikiMultiHopQA | O2 Searcher | 55.20 | 21.99 | 26.42 | 2.42 |
| | SimpleDeepSearcher | 60.50 | 17.97 | 24.38 | 3.64 |
| | R1 Searcher | 60.40 | 21.60 | 29.29 | 2.36 |
| | Search-R1 | 47.70 | 15.89 | 20.78 | 1.89 |
| | CoRAG | 65.40 | 22.79 | 30.67 | 4.00 |
| | **FrugalRAG (ours)** | 59.90 | 15.45 | 23.09 | 3.03 |
| MuSiQue | O2 Searcher | 37.00 | 14.25 | 21.37 | 1.95 |
| | SimpleDeepSearcher | 50.40 | 14.79 | 22.87 | 2.86 |
| | R1 Searcher | 57.70 | 15.77 | 24.77 | 2.31 |
| | Search-R1 | 38.10 | 12.65 | 19.86 | 1.36 |
| | CoRAG | 54.00 | 16.71 | 25.97 | 4.00 |
| | **FrugalRAG (ours)** | 52.60 | 15.13 | 23.96 | 3.30 |

Table 13: Retrieval metrics across datasets. We report Recall (Rec), Precision (Prec), F1, and average number of searches.

Table 14: FRUGALRAG demonstrates robust cross-dataset performance. The model is adapted using 1,000 training examples from the dataset shown in parentheses.

| Training Dataset | HotPotQA | | | 2Wiki | | | MuSiQue | | |
|---|---|---|---|---|---|---|---|---|---|
| | Recall | MBE | Searches | Recall | MBE | Searches | Recall | MBE | Searches |
| FRUGALRAG-7B + Qwen2.5-7B-Inst (E5) | | | | | | | | | |
| HotPotQA | 70.40 | 58.50 | 2.89 | 60.40 | 54.20 | 3.62 | 53.70 | 36.60 | 3.47 |
| 2Wiki | 71.00 | 59.30 | 2.85 | 58.80 | 50.40 | 3.03 | 52.22 | 36.10 | 3.38 |
| MuSiQue | 69.10 | 57.50 | 2.72 | 59.70 | 51.30 | 3.33 | 53.30 | 36.40 | 3.30 |
| FRUGALRAG-7B + Qwen2.5-32B-Inst (E5) | | | | | | | | | |
| HotPotQA | 70.40 | 61.40 | 2.89 | 60.40 | 54.20 | 3.62 | 53.70 | 37.60 | 3.47 |
| 2Wiki | 71.00 | 61.10 | 2.85 | 58.80 | 50.90 | 3.03 | 52.22 | 38.90 | 3.38 |
| MuSiQue | 69.10 | 60.10 | 2.72 | 59.70 | 52.40 | 3.33 | 53.30 | 38.90 | 3.30 |
| FRUGALRAG + Qwen2.5-7B-Inst (ColBERTv2) | | | | | | | | | |
| HotPotQA | 82.80 | 68.47 | 2.05 | 64.20 | 48.47 | 3.07 | 53.80 | 36.27 | 2.75 |
| 2Wiki | 81.90 | 68.34 | 2.56 | 63.50 | 48.93 | 2.95 | 53.80 | 34.30 | 3.10 |
| MuSiQue | 83.10 | 61.11 | 2.53 | 60.80 | 46.41 | 3.13 | 51.70 | 29.80 | 3.02 |

Hutter, 2017) with a learning rate of $2 \times 10^{-5}$. We use a linear learning rate scheduler with a warmup phase of 20 steps. The training is performed using 8 H100 80GB GPUs.

**Controlling test-time compute with RL.** Our RL step employs GRPO for fine-tuning the base policy $f_S$. Specifically, following the notation in DeepSeekMath (Shao et al., 2024), for each question $Q^{(j)}$, we sample a group of outputs $\{o_h^1, o_h^2, \ldots, o_h^v\}$ at hop $h$, where $v$ is set to 8. We optimize our base policy $f_S$ using the standard GRPO objective using the cumulative rollout reward as defined in Eq. 1. We use a KL divergence penalty with weight $0.1$ since we have a trained base policy, and set the maximum reward $R_{\max} = 2.0$ for stability. Generation is limited to a maximum of 256 completion tokens and the maximum prompt size is 1024. Training is conducted using DeepSpeed-Zero2 (Rasley et al., 2020) and 7 H100 GPUs (where 1 is exclusively reserved for sampling). We set the learning rate to $10^{-6}$. Due to the long prompt (which includes retrieved documents from previous hops), we use a total batch size of 48. We train FRUGALRAG for 400 steps across datasets and models, and report the performance using the final checkpoint.

**Algorithm 1:** Our novel two-stage framework, FRUGALRAG consists of **(1)** *Dataset Generation* and *Supervised "Explore" Finetuning*, and **(2)** *Controlling test-time compute wth RL*.

**Input:** Labeled dataset $\mathcal{L} = \{(Q^{(j)}, Y^{(j)})\}_{j=1}^{1000}$, $\mathcal{L}_{\text{init}} = \{(Q^{(j)}, Y^{(j)})\}_{j=1}^{50}$, retriever $\mathcal{R}$, base LM $f$, budget $B$, max hops $m$, number of samples $v$

    `// Prompt Optimization`
1  Perform prompt optimization on $f$ using $\mathcal{L}_{\text{init}}$ to obtain few-shot prompts $\{p_1, \ldots, p_n\}$;

    `// Dataset Generation`
2  Initialize finetuning dataset: $\mathbf{D} \leftarrow []$;
3  **for** $Q^{(j)}, Y^{(j)}$ *in* $\mathcal{L}$ **do**
4      Initialize buffer: `main_rollout` $\leftarrow []$;
5      Initialize $\mathcal{D}_0 \leftarrow \mathcal{R}(Q^{(j)})$ or $\emptyset$;
6      Initialize $T_0, A_0$;
7      $\mathcal{H}_0 \leftarrow \{Q^{(j)}, T_0, A_0, \mathcal{D}_0\}$ `// stores previous context`
8      Append $\mathcal{H}_0$ to `main_rollout`;
9      **for** $h = 1$ *to* $m$ **do**
10         **for** $i$ *in* $1 \ldots n$ **do**
11            **for** $h = 1$ *to* $B$ **do**
12               $(T_h^i, A_h^i, S_h^i) \leftarrow f(\mathcal{H}_{h-1}^i; p_i)$;
                    `// occurs in 10% of calls`
13               **if** $A_h^i = $ FINISH **then**
14                  **break**
15               $\mathcal{D}_h^i \leftarrow \mathcal{R}(S_h^i)$;
16               Remove duplicate retrievals from $\mathcal{D}_h^i$ ;
17               $\mathcal{H}_h^i \leftarrow \mathcal{H}_{h-1}^i \cup \{T_h^i, A_h^i, S_h^i, \mathcal{D}_h^i\}$;
18         Evaluate all $\{\mathcal{D}_h^i\}_{i=1}^n$ (recall against ground truth $Y^{(j)}$);
19         Select best-performing trajectory $\mathcal{H}^*$;
20         Append $\mathcal{H}^*$ to `main_rollout`;
21      Append each hop from `main_rollout` to $\mathbf{D}$;

    `// Stage 1: Supervised "Explore" Finetuning`
22  $f_S \leftarrow$ Fine-tune $f$ on $\mathbf{D}$ using standard next-token prediction `// See Eq. 2`

    `// Stage 2: Controlling test-time compute with RL`
23  **for** $Q^{(j)}, Y^{(j)}$ *in* $\mathcal{L}$ **do**
24      **for** $h = 1$ *to* $m$ **do**
25         Generate $v$ sample tuples $\{T_h^i, A_h^i, S_h^i, \mathcal{D}_h^i\}_{i=1}^v$ for hop $h$ ;
26      **for** $i = 1$ *to* $v$ **do**
27         Compute reward $R^i \leftarrow \mathbf{R}(\{\mathcal{D}_h^i\}_{h=1}^m, Y^{(j)}, f_S)$ `// See Eq. 1`
28         Backpropagate loss on $\{T_h^i, A_h^i, S_h^i\}_{h=1}^m$ using $R^i$;

| Question | Abbreviated Reasoning Trace | # Searches |
|---|---|---|
| **Who is the spouse of the screenwriter of *The Actress*?** | **T0:** Search original question. 
 **T1:** Initial results mention unrelated screenwriters (e.g., David Fury); refine query. 
 **T2:** Search "screenwriter of The Actress movie". 
 **T3:** Identify screenwriter as Ruth Gordon. 
 **T4:** Search "Ruth Gordon spouse" → Garson Kanin. | 4 |
| **What administrative territorial entity is the owner of Ciudad Deportiva located?** | **T0:** Search original question. 
 **T1:** Identify location: Benito Juárez borough, Mexico City. 
 **T2:** Refine query to determine owning administrative entity → Local government of Madrid. | 2 |
| **In which borough was Callum McManaman born?** | **T0:** Search original question. 
 **T1:** Retrieve birthplace: Huyton → Borough of Knowsley. | 1 |

Table 15: Qualitative reasoning traces demonstrating adaptive retrieval depth. More complex multi-hop questions require iterative query refinement and additional searches, whereas simpler factual queries terminate early.

## G  DATASET AND RETRIEVAL INDEX

We use the pre-processed Wikipedia abstracts index [1] provided by ColBERTv2 (Santhanam et al., 2021) for all our experiments on HotPotQA (Yang et al., 2018). For each instance, we retrieve the top 3 documents and their titles and perform a maximum 6 retrievals. HotPotQA annotations consists of document title and evidence sentences which are used to compute the Recall and Supporting Document F1 respectively.

Since 2WikiMultiHopQA (Ho et al., 2020) and MuSiQue (Trivedi et al., 2022b) datasets are created using both the body and abstract of wikipedia articles we use the pre-processed dump of Wikipedia provided by Karpukhin et al. (2020) and index it using ColBERTv2 (Santhanam et al., 2021). The generated index consists of 21M passages. For each instance, we retrieve top 5 documents and append it to our context. For experiments in Table 2, we use E5-base provided by FlashRAG Jin et al. (2025b) indexed on Wikipedia Petroni et al. (2021) which consists of 21 million passages, and retrieve top 5 documents for all datasets.

## H  PROMPT DETAILS

For completeness, we provide the exact prompt templates used for the ReAct-style reasoner model and the answer extraction model which utilize DsPy (Khattab et al., 2023). Note that after prompt optimization, the demos field is populated but is omitted here for brevity. Additionally, we provide the instruction used for computing MBE using FlashRAG (Jin et al., 2025b) prompt.

**Prompt (with FINISH).** React is the reasoner model prompt and extract is answer model prompt.

```
{
 "react": {
  "signature": {
   "instructions": "Given the fields `question`, produce the fields `
      answer`.\n\nYou will be given `question` and your goal is to
      finish with `answer`.\n\nTo do this, you will interleave Thought,
       Tool Name, and Tool Args, and receive a resulting Observation.\n
      \nThought can reason about the current situation, and Tool Name
      can be the following types:\n\n(1) AdvancedSearch, whose
      description is <desc>Searches documents using a search_query.
      Arguments: - \"search_query\": an optimized search query for
```

---

[1] `https://downloads.cs.stanford.edu/nlp/data/colbert/baleen/wiki.abstracts.2017.tar.gz`

```
                  dense passage retrieval. IMPORTANT: YOU MUST always PROVIDE \"
                  search_query\" in the arguments!</desc>. It takes arguments {'
                  search_query': 'str', 'trajectory_id': 'Union[str, NoneType]'} in
                   JSON format.\n(2) finish, whose description is <desc>Signals
                  that the final outputs, i.e. 'answer', are now available and
                  marks the task as complete.</desc>. It takes arguments {} in JSON
                   format.",
        "fields": [
          {"prefix": "Question:", "description": "${question}"},
          {"prefix": "Trajectory:", "description": "${trajectory}"},
          {"prefix": "Next Thought:", "description": "${next_thought}"},
          {"prefix": "Next Tool Name:", "description": "${next_tool_name}"},
          {"prefix": "Next Tool Args:", "description": "${next_tool_args}"}
        ]
      }
    },
    "extract": {
      "signature": {
        "instructions": "Given the fields 'question', produce the fields '
            answer'.",
        "fields": [
          {"prefix": "Question:", "description": "${question}"},
          {"prefix": "Trajectory:", "description": "${trajectory}"},
          {"prefix": "Reasoning: Let's think step by step in order to", "
              description": "${reasoning}"},
          {"prefix": "Answer:", "description": "${answer}"}
        ]
      }
    }
  }
}
```

**Prompt (without FINISH).**

```
{
  "react": {
    "signature": {
      "instructions": "Given the fields 'question', produce the fields '
          answer'.

You will be given 'question'.

You must decide what to do next by providing:

- Thought: reasoning about the current step
- Tool Name: one of the available tools
- Tool Args: A valid JSON with the necessary keys.

Repeat this for a few steps to gather information.

You do not need to produce a final answer - just use the tools
    iteratively.
Make sure to include all required Tool Args as a valid JSON object.

(1) AdvancedSearch: Searches documents using a search_query.
Arguments:
 - \"search_query\": an optimized search query for dense passage
     retrieval.
 - \"trajectory_id\": optional string.
IMPORTANT: YOU MUST always PROVIDE \"search_query\"."
    ,
    "fields": [
      {"prefix": "Question:", "description": "${question}"},
      {"prefix": "Trajectory:", "description": "${trajectory}"},
      {"prefix": "Next Thought:", "description": "${next_thought}"},
      {"prefix": "Next Tool Name:", "description": "${next_tool_name}"},
```

```
        {"prefix": "Next Tool Args:", "description": "${next_tool_args}"}
      ]
    }
  },
  "extract": {
    "signature": {
      "instructions": "Given the fields 'question', produce the fields '
          answer'.",
      "fields": [
        {"prefix": "Question:", "description": "${question}"},
        {"prefix": "Trajectory:", "description": "${trajectory}"},
        {"prefix": "Reasoning: Let's think step by step in order to", "
            description": "${reasoning}"},
        {"prefix": "Answer:", "description": "${answer}"}
      ]
    }
  }
}
```

**Answer extraction for E5.** For the E5 results, we use a simpler alternative answer extraction mechanism to ensure that the answer extraction is less noisy.

```
Answer the following question using ONLY the provided documents.

Question: ${question}

## Retrieved Documents
${doc_context}

Think step by step, then give your final answer.
```

**MBE Prompt.**

```
Judge whether the following [response] to [question] is correct or not
    based on the precise and unambiguous [correct_answer] below.

[question]: {question}

[response]: {response}

[correct_answer]: {correct_answer}

Your judgement must be in the format and criteria specified below:

extracted_final_answer: The final exact answer extracted from the [
    response].

[correct_answer]: Repeat the [correct_answer] given above.

reasoning: Explain why the extracted_final_answer is correct or incorrect
     based on [correct_answer], in the context of this [question]. You
    should judge whether the extracted_final_answer is semantically
    equivalent to [correct_answer], allowing the extracted_final_answer
    to be string variations of [correct_answer]. You should also allow
    the extracted_final_answer to be more precise or verbose than [
    correct_answer], as long as its additional details are correct. Do
    not comment on any background to the problem, do not attempt to solve
     the problem, do not argue for any answer different than [
    correct_answer], focus only on whether the answers are semantically
    equivalent.

correct: Answer 'yes' if extracted_final_answer matches the [
    correct_answer] given above. Answer 'no' otherwise, i.e. if there if
    there is any inconsistency, ambiguity, non-equivalency, or if the
```

```
    extracted answer is incorrect. If the response lists multiple items
    and one of them matches the correct answer, judge as correct UNLESS
    the question specifically asks for a single item and the additional
    items are wrong.

confidence: The extracted confidence score between 0|\%| and 100|\%| from
    [response]. Put 100 if there is no confidence score available.
```

