# OpenReview forum: "FrugalRAG: Less is More in RL Finetuning for Multi-hop Question Answering"
_ICLR.cc/2026/Conference — ICLR 2026 Poster_

### Official Review · Reviewer_9twQ · 2025-10-28

**Soundness:** 2
**Presentation:** 3
**Contribution:** 2
**Rating:** 4
**Confidence:** 4

**Summary:**

This paper introduces FrugalRAG, a two-stage finetuning framework that combines supervised exploration and reinforcement learning to optimize retrieval efficiency in multi-hop question answering. Instead of simply scaling test-time retrieval, the approach adaptively determines the number of retrieval steps per query, balancing accuracy with efficiency. Experiments across benchmarks such as HotPotQA, 2Wiki, MuSiQue, and BrowseCompPlus show that FrugalRAG achieves strong efficiency–accuracy tradeoffs while training on only 1,000 examples.

**Strengths:**

1. The focus on frugality addresses a critical yet underexplored dimension in RAG systems (efficiency, not just accuracy).

2. The proposed method achieves competitive or superior results with only 1,000 training examples is impressive compared to prior work requiring 100k+.

**Weaknesses:**

1. The comparison is unfair by natural, as baseline uses weak retriever (not colbert v2).

2. The RL reward design assumes access to the “optimal” stopping point based on ground-truth evidence, which is often unavailable or impractical in real-world applications.

3. While the paper emphasizes data efficiency and search efficiency as central contributions, these concepts have been discussed extensively in prior work (e.g., [1,2,3, 4]). The paper would benefit from a clearer articulation of how its framing or results go beyond existing definitions of efficiency.

4. Limited case studies and inference efficiency studies of the proposed method.


[1] Wang, Hongru, Cheng Qian, Wanjun Zhong, Xiusi Chen, Jiahao Qiu, Shijue Huang, Bowen Jin, Mengdi Wang, Kam-Fai Wong, and Heng Ji. "Acting Less is Reasoning More! Teaching Model to Act Efficiently." arXiv preprint arXiv:2504.14870 (2025).

[2] Jiang, Pengcheng, Xueqiang Xu, Jiacheng Lin, Jinfeng Xiao, Zifeng Wang, Jimeng Sun, and Jiawei Han. "s3: You Don't Need That Much Data to Train a Search Agent via RL." arXiv preprint arXiv:2505.14146 (2025).

[3] Li, Zhuofeng, Haoxiang Zhang, Seungju Han, Sheng Liu, Jianwen Xie, Yu Zhang, Yejin Choi, James Zou, and Pan Lu. "In-the-Flow Agentic System Optimization for Effective Planning and Tool Use." arXiv preprint arXiv:2510.05592 (2025).

[4] Song, Huatong, Jinhao Jiang, Wenqing Tian, Zhipeng Chen, Yuhuan Wu, Jiahao Zhao, Yingqian Min, Wayne Xin Zhao, Lei Fang, and Ji-Rong Wen. "R1-Searcher++: Incentivizing the Dynamic Knowledge Acquisition of LLMs via Reinforcement Learning." arXiv preprint arXiv:2505.17005 (2025).

**Questions:**

Suggestion: The exposition is sometimes dense, with technical details (e.g., rollout generation, reward shaping) overshadowing high-level intuition. A clearer articulation of practical implications and limitations would improve readability.


Could you provide more detailed analysis or case studies showing how the system behaves on questions of varying complexity?

Since the method uses only 1,000 labeled examples, how sensitive is performance to the exact number of examples? Have you tested smaller budgets (e.g., 100 or 500 examples) to see the degradation curve? I am wondering what is the performance of the baseline with the same amount of number of examples.

---

> ### Author Response · Authors · 2025-11-21
> **Rebuttal Part 1**
>
> We sincerely thank you for your efforts and time. Below we address each of the concerns raised.
>
> **Unfair Comparison.** All comparisons in Table 2 use exactly the same retriever, index, and evaluator. As stated in Line 320, every method compared in Table 2, including Search-R1 and CoRAG, is evaluated under the FlashRAG setup (Jin et al., 2025). FlashRAG standardizes the retriever to E5-base-v2, and Index to Wikipedia 21M. We do not mix retrievers or indices within any row of Table 2. This ensures strict apples-to-apples comparisons. Table 1’s purpose is to validate the effectiveness of FrugalRAG using a different retriever (ColbertV2). We do not draw cross-paper causal conclusions from Table 1 (all comparisons are confined to Table 2); this evaluation simply demonstrates that FrugalRAG remains effective even with a different retrievers. Across all evaluations, we use the Qwen-32B judge, following the BrowseComp-Plus. Prompt templates are also identical for all methods.
>
> **Ground Truth evidence.** Our paper argues that, unlike prior work requiring large training sets and using only the final answer as reward, a small number of evidence-recall rewards can serve as an effective alternative. We emphasize that relying on these signals is an intentional design choice rather than a limitation, and it enables efficient training with far fewer examples as demonstrated in our results. Our motivation stems from open ended QA on private data or rapidly changing data (e.g., new events). Instead of training a model to optimize final answer accuracy (where many of the answers may be inferred from parametric memory), we train the  model to search for the right documents efficiently, not just the final answer. We agree that this may be limited availability, that's why we show that only 1000 of such dense rewards are needed. In another ablation (Table 10), we show how our algorithm works even if 500 examples are given. This number can be curated easily by organizations who have a specific RAG usecase.
>
> **Related works.** Thank you for pointing out these works. We would like to highlight that some of these works are contemporaneous (Jiang et al. EMNLP 2025, Song et al. EMNLP 2025) and others are either unpublished and were **made public after the ICLR deadline** (Wang et al. May 2025, Li et al. Oct 2025).
>
> Wang et al. (Arxiv, May 2025) focus on improving tool-calling productivity by introducing a decaying reward to discourage longer trajectories. Jiang et al. (EMNLP, Nov 2025) propose s3, a data-efficient framework that trains the reasoner separately while using an LLM as a generator, showing that relying solely on answer-based rewards can cause overfitting. This *complements our work*, highlighting the shortcomings of answer-based rewards. Li et al. (Oct 2025) present a modular agentic system that optimizes the planner/reasoner using verifiable rewards applied only after multiple turns. R1-Searcher++ (May 2025) adopts a two-stage pipeline, SFT initialization followed by RL with multiple reward signals (answer, group-level, and format) but does not focus on stopping criterion.
>
> Our work differs from all of these methods. FrugalRAG emphasizes (1) the explicit role of exploration in stage 1 and (2) a reformulation of standard answer-based rewards into "learning when to stop". This reframing directly addresses the core inefficiency in many search-based agents, both in finetuning and inference. To our knowledge, prior work has not used this framing as an explicit learning signal.
>
> **Performance on varying scales.** We expanded our training data using HotpotQA and varied the size across 100, 500, 1000 (our original setting), and 5000 examples. We evaluate the resulting models across three datasets (HotpotQA, 2Wiki, Musique).
>
> Table 14: Sample Count Ablation
> | **Dataset**  | **Metric** | **100** | **500** | **1000** | **5000** |
> | ------------ | ---------- | ------- | ------- | -------- | -------- |
> | **HotpotQA** | Recall     | 67.9    | 70.8    | 70.4     | 69.1     |
> |              | Searches   | 2.873   | 3.243   | 2.89     | 2.36     |
> | **2Wiki**    | Recall     | 58.5    | 59.1    | 60.04    | 59.3     |
> |              | Searches   | 3.54    | 3.79    | 3.623    | 3.01     |
> | **MuSiQue**  | Recall     | 51.6    | 53.6    | 53.7     | 52.4     |
> |              | Searches   | 3.532   | 3.774   | 3.47     | 3.038    |
>
> Across all datasets, performance increases substantially from 100 to 500 examples. For example, from 100 to 500 examples, the recall improves from 67.9 to 70.80 on HotpotQA, and the same trend appears in 2Wiki and Musique. This shows that our method quickly learns the core reasoning required for retrieval with only a few hundred supervised examples. Interestingly, the search efficiency stabilizes with more data (with 5000 examples searches drop consistently across the three datasets). This substantiates that with increasing number of examples our method improves the overall performance.

---

> ### Author Response · Authors · 2025-11-21
> **Rebuttal Part 2**
>
> **Case studies.** In Table 5 of the main paper, we show that with increasing difficulty, FrugalRAG on average adaptively issues more queries. In addition to this, in Figure 2, we show that this trend is significant across a large number of examples on both 2WikiMultiHopQA and MuSiQue. Additionally we conduct additional rebuttal experiments on efficiency which are presented in Tables 2, 3, 4 (reviewer: GAQh).  In our revised manuscript we will include qualitative examples to substantiate our claim. Below, we provide qualitative examples on MuSiQue:
>
> ```Example 1 -
> "Who is the spouse of the screenwriter of The Actress?"
>
> Reasoning Trace:
> "Thought 0": "I will start by searching for relevant documents with the original question as the search query."
> "Search Query 0": "Who is the spouse of the screenwriter of The Actress?"
> "Thought 1": "From the search results, I see that David Fury is a screenwriter, but there is no mention of \"The Actress\". I need to refine my search to find the screenwriter of \"The Actress\" and then find their spouse."
> "Search Query 1": "screenwriter of The Actress"
> "Thought 2": "I need to find the screenwriter of \"The Actress\" and then find their spouse."
> "Search Query 2": "screenwriter of The Actress movie"
> "Thought 3": "The screenwriter of \"The Actress\" movie is Ruth Gordon. Now I need to find her spouse."
> "Search Query 3": "Ruth Gordon spouse"
> "Thought 4": "The screenwriter of \"The Actress\" is Ruth Gordon, and her spouse was Garson Kanin."
> ```
>
> ```Example 2 -
> "What administrative territorial entity is the owner of Ciudad Deportiva located?"
>
> Reasoning Trace:
> "Thought 0": I will start by searching for relevant documents with the original question as the search query.
> "Search Query 0": "What administrative territorial entity is the owner of Ciudad Deportiva located?"
> "Thought 1": "From the search results, I see that Ciudad Deportiva is located in the Benito Ju\u00e1rez borough of Mexico City. Now, I need to determine the administrative territorial entity that owns Ciudad Deportiva."
> "Search Query 1": "administrative territorial entity owner of Ciudad Deportiva"
> "Thought 2": "The search results indicate that the administrative territorial entity owning Ciudad Deportiva is the local government of Madrid. I can now provide the answer to the question."
> ```
>
> ```Example 3 -
> "In which borough was Callum McManaman born?"
>
> Reasoning Trace:
> "Thought 0": "I will start by searching for relevant documents with the original question as the search query."
> "Search Query 0": "In which borough was Callum McManaman born?"
> "Thought 1": "From the search results, I can see that Callum McManaman was born in Huyton, which is part of the borough of Knowsley. I will now provide the answer to the user's question."
> ```
>
> The examples above demonstrate that the model is able to issue a larger number of search queries for more challenging questions in the MuSiQue dataset. We will include these examples in the revised version of the manuscript.
>
> **Additional baselines** In Table 2 of the main paper, we already provide several baseline comparisons. Under the same 1000-example training budget, Ret-Robust performs substantially worse than FrugalRAG across all three datasets. To further isolate the effect of our method under this low-data regime, we conducted additional ablations using 1000 training examples, summarized below:
>
> Table 15: Additional Ablations:
> |  Baseline (HotpotQA)                  | Recall    | Searches |
> | -------------------------------------------- | --------- | -------- |
> | Qwen2.5-7B-Instruct + Recall Reward          | 66.8      | 2.57     |
> | FrugalRAG-Explore w/ Recall Reward            | 76.0      | 6.00     |
> | Stage-2 without Stage-1 (w/o SFT)           | 66.5      | 2.36     |
> | **FrugalRAG (Ours)**                         | 70.40 | 2.89 |
>
> Recall-only reward is insufficient: Training Qwen2.5-7B-Instruct directly with a Recall reward yields significantly worse performance, indicating that naively applying RL does not match FrugalRAG.
>
> Our two-stage design is necessary: Removing Stage-1 (i.e., training Stage-2 RL from scratch) degrades both recall and retrieval behavior.

---

### Official Review · Reviewer_r6Km · 2025-10-30

**Soundness:** 2
**Presentation:** 3
**Contribution:** 2
**Rating:** 4
**Confidence:** 5

**Summary:**

The paper proposes FrugalRAG, a two-stage finetuning framework for multi-hop RAG: Stage-1 supervised “explore” training to maximize evidence coverage, followed by Stage-2 RL to adaptively stop retrieval, balancing answer quality against search cost. With ~1k training examples, the method reports strong efficiency–accuracy tradeoffs and adaptive compute (e.g., fewer searches on easier questions) across HotPotQA, 2Wiki, MuSiQue, and some generalization to BrowseCompPlus.

**Strengths:**

1. Separating exploration (coverage) from stopping (efficiency) is a clean, actionable design; the reward explicitly ties termination to an “optimal” hop length and format compliance.

2. Tables show fewer searches at comparable or better MBE/recall versus strong prompting and SFT baselines, and the “tradeoff” metric highlights practical benefits.

3. The method increases search steps with question difficulty and shows some cross-dataset transfer; the modularity across retrievers/generators is also demonstrated.

**Weaknesses:**

1. Motivation around “less data” is under-argued: The paper asserts success with ~1k examples, but the core motivation, why low-label regimes are critical for RAG, feels underspecified. In many RAG settings, labeled QA pairs or weak labels are not rare (e.g., synthetic or distantly-supervised pipelines), and the paper does not clarify whether performance keeps improving with more data (scaling behavior is central to RAG systems). A scaling curve (1k→10k→100k) would make this claim concrete.

2. Limited novelty. Algorithmically, Stage-2 introduces a termination-oriented reward shaping that discourages extra retrieval steps; this reads as a targeted constraint atop existing RL-for-RAG methods rather than a fundamentally new learning principle. The contribution is useful but incremental.

3. Missing ablation on the retrieval-steps component of the reward. Since the central claim is “RL learns to stop,” an ablation that removes the step-penalty/termination reward is necessary to show the gain is not just from better exploration traces or general RL regularization. (Table/figures do not isolate this factor.)

4. Questionable cross-method comparisons due to retriever mismatch. Several headline comparisons juxtapose methods that use different retrievers (e.g., E5-base-v2 vs. ColBERTv2) or different indices/evaluators. This undermines causal conclusions about the policy’s quality. A strict apples-to-apples study (same retriever, index, judge) is needed for Search-R1/CoRAG comparisons.

5. Related work coverage is incomplete. Important recent efforts are missing or not discussed in sufficient depth, such as [1-3].

[1] Wang X, Wang Z, Gao X, et al. Searching for best practices in retrieval-augmented generation[J]. arXiv preprint arXiv:2407.01219, 2024.

[2] Ammann P J L, Golde J, Akbik A. Question Decomposition for Retrieval-Augmented Generation[J]. arXiv preprint arXiv:2507.00355, 2025.

[3] Xu R, Shi W, Zhuang Y, et al. Collab-rag: Boosting retrieval-augmented generation for complex question answering via white-box and black-box llm collaboration[J]. arXiv preprint arXiv:2504.04915, 2025.

**Questions:**

refer to weakness

---

> ### Author Response · Authors · 2025-11-21
> **Rebuttal Part 1**
>
> Thank you for your thoughtful feedback.
>
> **Less data motivation.** Our motivation stems from open ended QA on private data or rapidly changing data (e.g., new events). Instead of training a model to optimize final answer accuracy (where many of the answers may be inferred from parametric memory), we train the  model to search for the right documents efficiently, not just the final answer.
>
> As also requested by reviewer GAQh, we conducted additional experiments to study the scaling behavior of our approach. While running full scaling curves is computationally expensive, we expanded our training data using HotpotQA and varied the size across 100, 500, 1000 (our original setting), and 5000 examples. We evaluate the resulting models across three datasets (HotpotQA, 2Wiki, Musique).
>
> Table 12: Sample Count Ablation
> | **Dataset**  | **Metric** | **100** | **500** | **1000** | **5000** |
> | ------------ | ---------- | ------- | ------- | -------- | -------- |
> | **HotpotQA** | Recall     | 67.9    | 70.8    | 70.4     | 69.1     |
> |              | Searches   | 2.873   | 3.243   | 2.89     | 2.36     |
> | **2Wiki**    | Recall     | 58.5    | 59.1    | 60.04    | 59.3     |
> |              | Searches   | 3.54    | 3.79    | 3.623    | 3.01     |
> | **MuSiQue**  | Recall     | 51.6    | 53.6    | 53.7     | 52.4     |
> |              | Searches   | 3.532   | 3.774   | 3.47     | 3.038    |
>
> Across all datasets, performance increases substantially from 100 to 500 examples. For example, from 100 to 500 examples, the recall improves from 67.9 to 70.80 on HotpotQA, and the same trend appears in 2Wiki and Musique. This shows that our method quickly learns the core reasoning required for retrieval with only a few hundred supervised examples. Interestingly, the search efficiency stabilizes with more data (with 5000 examples searches drop consistently across the three datasets). This substantiates that with increasing number of examples our method improves the overall performance. We are currently running more experiments with 10k examples and will share those results once completed.
>
> **Novelty.** Thank you for your comment. We respectfully disagree and argue that our contributions are novel. Prior RL-for-RAG methods overwhelmingly adopt math/code-style RL they reward only final-answer correctness and rely on a large number of traces (e.g., Search-R1, R1-Searcher, etc). As we show in the paper, directly using this paradigm in multi-hop RAG is not trivial as it leads to inefficient training and inference. Our formulation transforms the problem to learn when to stop retrieving more information. The two-stage curriculum targetted for efficiency leveraging only 1000 examples does not appear in prior literature. Overall, novelty lies in problem formulation, curriculum design, and the new role of RL in multi-hop RAG.
>
> **Comparison Fairness.** All headline comparisons in Table 2 use exactly the same retriever, index, and evaluator. As stated in Line 320, every method compared in Table 2, including Search-R1 and CoRAG, is evaluated under the FlashRAG setup (Jin et al., 2025). FlashRAG standardizes the retriever to E5-base-v2, and Index to Wikipedia 21M. We do not mix retrievers or indices within any row of Table 2. This ensures strict apples-to-apples comparisons.
>
> Table 1’s purpose is to validate the effectiveness of FrugalRAG using a different retriever (ColbertV2). We do not draw cross-paper causal conclusions from Table 1 (all comparisons are confined to Table 2). Across all evaluations, we use the Qwen-32B judge, following the BrowseComp-Plus. Prompt templates are also identical for all methods.

---

> ### Author Response · Authors · 2025-11-21
> **Rebuttal Part 2**
>
> **Additional Ablations.** We thank the reviewer for their suggestion. As requested, we conduct additional ablations to isolate the effect of our reward design.
>
> Table 13: Additional ablations on reward design
> | Ablation Variant (HotpotQA)                  | Recall    | Searches |
> | -------------------------------------------- | --------- | -------- |
> | FrugalRAG w/o Late Penalty                   | 70.5      | 3.144    |
> | Stage-2 without Stage-1 (w/o SFT)           | 66.5      | 2.36     |
> | **FrugalRAG (Ours)**                         | **70.40** | **2.89** |
>
> Specifically, we present results with and without the late penalty and stage 1 finetuning. Our results demonstrate that FrugalRAG outperforms both these variants. Removing the penalty term results in larger number of searches compared to FrugalRAG and not performing SFT results in lower recall. These findings substantiate that our reward design is helpful and not merely the result of regularization. We will include these results in the updated manuscript.
>
> **Related Work coverage.** Thank you for highlighting these recent works. We appreciate the opportunity to clarify how they relate to our contributions and will incorporate them into the final revision. Wang et al. provide a comprehensive empirical study of best practices for RAG, focusing on retrieval architectures, negative sampling strategies, and reranking. FrugalRAG contributes a new RL-based framework for dynamically adjusting retrieval depth which is beyond the scope of Wang et al. Ammann et al. introduce a question-decomposition and reranking system for multihop RAG. Their approach improves retrieval quality through decomposition, whereas our approach focuses primarly on learning to adaptively retrieve. Xu et al. propose Collab-RAG, leveraging collaboration between an LLM and an SLM. This method relies on coordination between models, whereas FrugalRAG operates entirely with SLM only. We will make these distinctions clear in our revised manuscript.

---

### Official Review · Reviewer_2odc · 2025-10-30

**Soundness:** 3
**Presentation:** 3
**Contribution:** 4
**Rating:** 6
**Confidence:** 4

**Summary:**

The paper proposes FrugalRAG, a two-stage finetuning method that makes language models better at multi-hop question answering with retrieval-augmented generation while staying efficient. First, it uses supervised finetuning on about 1,000 examples to teach the model how to write good search queries and gather relevant evidence across multiple hops. Then it uses reinforcement learning not to make the model search more, but to teach it when to stop searching once it has enough information. This lets the model adapt its compute: easy questions get only a couple searches, hard questions get more. FrugalRAG matches or beats prior systems on benchmarks like HotPotQA in both answer accuracy and evidence recall, while using fewer retrieval calls and far less training data than existing RL approaches.

**Strengths:**

S1. Clear problem focus. The paper presents a well-defined motivation by targeting the inefficiency problem in multi-hop RAG systems, where existing RL/RAG agents often over-retrieve and waste searches. FrugalRAG directly addresses this issue by introducing an adaptive stopping mechanism that halts retrieval once sufficient evidence is gathered, ensuring both cost efficiency and effectiveness.

S2. Logical two-stage design. The technical structure is intuitive and easy to follow.
Stage 1 uses supervised fine-tuning to learn effective retrieval, and Stage 2 applies RL to decide when to stop searching—cleanly separating exploration and stopping behaviors.

S3. Strong efficiency and quality gains. The method achieves notable efficiency improvements while maintaining or improving accuracy.

**Weaknesses:**

W1. Concerns about technical design

(1) Line 251 defines “$\Delta$ is the normalized difference $(h_\text{term} - h) / B$”, but only $h_\text{term}$, $h*$, and $B$ are defined before. Should it be “$\Delta$ = $(h_\text{term} - h) / B$”? This matches the idea that the penalty depends on how far you are from the optimal stopping hop.

(2) Lines 243–249 define a piecewise reward $R$ for late stop, perfect stop, and early stop. The “early stop” case is triggered by $c < \tau$ (low answer quality), and it reuses the same penalty term $log((1-\Delta)/\Delta)$. But if you stop early then $h_\text{term} < h*$, so $\Delta < 0$, and $log((1-\Delta)/\Delta)$ is undefined for negative $\Delta$ (the fraction becomes negative). As written, the early-stop reward cannot actually be evaluated, so the reward function is internally inconsistent.

(3) In Algorithm 1 (Line 766), the notation alternates between “budget $B$, max hops $m$”. Both $B$ and $m$ appear to mean “max hops”, which is confusing and should be unified.

(4) Around Line 237, the score $c$` and the threshold $\tau$ are both used in the reward but never clearly defined.

W2. Limited efficiency analysis

The paper reports “average searches per question” as its main efficiency metric, but does not provide end-to-end latency or monetary cost in realistic settings (e.g., retrieval time over a large index, API/tool-call pricing). For real RAG systems, wall-clock delay and dollars per query matter, and that systems-level cost analysis is missing.

W3. Missing ablations on RL design

We see that “FrugalRAG-Explore (SFT)” vs “FrugalRAG (RL)” works, but we do not get deeper ablations, such as: (i) alternative stopping rewards, (ii) RL that also tunes query generation instead of only stopping, or (iii) RL from scratch without Stage 1 supervision (iv) training with backbone pipelines other than Qwen2.5-7B-Instruct.

W4. Code release

There is no released code and no clear commitment to release it.

W5. Evaluation metric concerns

Answer quality is judged mainly with a model-based evaluator (“MBE”) rather than exact match or human review. That means reported gains could reflect the judging model’s stylistic preferences instead of true factual correctness. For retrieval, the paper mostly reports recall and does not report precision, F1, MAP, or MRR.

W6. Grammar issues

There are several grammar issues. Examples:

(1) Line 230: “our only goal is to learn when to sufficient evidence has been gathered.”

(2) Lines 52–53: a long sentence ending with “(with potentially private documents), Typical solutions…” incorrectly capitalizes “Typical” after a comma and fuses two sentences.

(3) Line 54: “train on 90,000-1,00,000 examples...” mixes numbering styles; it should be “90,000–100,000.”

**Questions:**

Q1. Could you explain the technical concerns in W1?
Q2. Do you have other cost measurements (e.g., wall-clock time, API/tool-call cost)?
Q3. Do you have more RL ablation studies?
Q4. How stable is RL training across random seeds and hyperparameters?
Q5. Will you release the code (especially for the RL loop and reward)?
Q6. Do you have other evaluation results such as F1, precision@k, MRR, or MAP?
Q7. How would you make the stopping policy safer in high-stakes settings where stopping too early and hallucinating is unacceptable?

---

> ### Author Response · Authors · 2025-11-21
> **Rebuttal Part 1**
>
> We sincerely thank you for your detailed feedback.
>
> **Equation 1.** Your observations are correct. In our code, we use $\\Delta$ as the absolute value of the normalized difference between the optimal stopping point $h*$ and the actual stopping point $h_{\\text{term}}$; $\\Delta = |h* - h_{\\text{term}}| / B$. When $c < \\tau$, we set $h* = B$, the maximum budget, ensuring that $\\log$ is not undefined; $\\log((1 - \\Delta) / \\Delta)$. We will clearly define all variables and include this correction in the revised version.
>
>
> **Notation.** Apologies for the inconsistencies, we will update our manuscript to use $B$ for budget everywhere.
>
> **Defining c, $\\tau$.** In Section *Optimal Rollout Length*, we explain how we obtain the example-specific threshold $\\tau$ using the FrugalRAG-Explore model. The threshold $\tau$ corresponds to the maximum recall score that the stage-1 model can achieve on that training example. During RL training, if the model’s current score falls below $\tau$, we incentivize it to explore further. We will revise the paper with detailed explanation.
>
> **Additional Metrics.** We computed FLOPs, tokens processed, and latency. FrugalRAG is competitive on compute-normalized metrics like tokens, FLOPs, and searches across datasets. We note that in practical deployments monetary cost is typically dominated by the number of searches and tokens, on which FrugalRAG is directly comparable to baselines.
>
> Table 2: Efficiency Comparison on HotPotQA
>
> | **Method**             | **Avg Tokens** | **FLOPs**              | **Latency (s)** | **Search Count** | **Recall (%)** |
> | ---------------------- | -------------- | ---------------------- | --------------- | ---------------- | -------------- |
> | **O2 Searcher-3B**        | 2,937          | $1.79 \times 10^{13}$  | 0.1050          | 1.77             | 50.10          |
> | **SimpleDeepSearcher-7B** | 9,657          | $1.31 \times 10^{14}$  | 0.6571          | 2.75             | 64.80          |
> | **R1 Searcher-7B**        | 4,458          | $6.17 \times 10^{13}$  | 0.1335          | 2.22             | 69.10          |
> | **Search-R1-7B**          | 2,119          | $2.99 \times 10^{13}$  | 0.0650          | 1.28             | 48.20          |
> | **CoRAG-8B**              | 7,770          | $1.24 \times 10^{14}$  | 0.1486          | 4.00             | 64.30          |
> | **FrugalRAG-7B (ours)**   | 9,138      | $1.27 \times 10^{14}$ | 0.2415          | 2.89             | 70.40          |
>
>
> Table 3: Efficiency Comparison on 2WikiMultiHopQA
> | **Method**             | **Tokens** | **FLOPs**              | **Latency (s)** | **Search Count** | **Recall (%)** |
> | ---------------------- | ---------- | ---------------------- | --------------- | ---------------- | -------------- |
> | **O2 Searcher-3B**        | 4,212      | $2.53 \times 10^{13}$  | 0.1206          | 2.42             | 55.20          |
> | **SimpleDeepSearcher-7B** | 13,433     | $1.80 \times 10^{14}$  | 0.7625          | 3.64             | 60.50          |
> | **R1 Searcher-7B**        | 4,939      | $6.89 \times 10^{13}$  | 0.1374          | 2.36             | 60.40          |
> | **Search-R1-7B**          | 3,456      | $4.96 \times 10^{13}$  | 0.0852          | 1.89             | 47.70          |
> | **CoRAG-8B**              | 8,018      | $1.28 \times 10^{14}$  | 0.1511          | 4.00             | 65.40          |
> | **FrugalRAG-7B (ours)**   | 11279      | $1.45 \times 10^{14}$ | 0.4673          | 3.03             | 59.90          |
>
>
> Table 4: Efficiency Comparison on MuSiQue
> | **Method**             | **Tokens** | **FLOPs**              | **Latency (s)** | **Search Count** | **Recall (%)** |
> | ---------------------- | ---------- | ---------------------- | --------------- | ---------------- | -------------- |
> | **O2 Searcher-3B**        | 2,923      | $1.74 \times 10^{13}$  | 0.1009          | 1.95             | 37.00          |
> | **SimpleDeepSearcher-7B** | 10,027     | $1.35 \times 10^{14}$  | 0.7366          | 2.86             | 50.40          |
> | **R1 Searcher-7B**        | 4,665      | $6.47 \times 10^{13}$  | 0.1327          | 2.31             | 57.70          |
> | **Search-R1-7B**          | 2,212      | $3.14 \times 10^{13}$  | 0.0676          | 1.36             | 38.10          |
> | **CoRAG-8B**              | 7,683      | $1.23 \times 10^{14}$  | 0.1472          | 4.00             | 54.00          |
> | **FrugalRAG-7B (ours)**   | 11914     | $1.66 \times 10^{14}$ | 0.4098          | 3.30             | 52.60          |

---

> ### Author Response · Authors · 2025-11-21
> **Rebuttal Part 2**
>
> **Additional Ablations.** Following your feedback, we conducted several additional ablations on the reward design to establish the efficacy of FrugalRAG. The table below summarizes our results:
>
> Table 7a: Comparison — w/o Stage-1 vs. Ours
>
> | Ablation                              | Recall | Searches |
> | ------------------------------------- | ------ | -------- |
> | Stage-2 without Stage-1 (w/o SFT)     | 66.5   | 2.36     |
> | **FrugalRAG (Ours)**                  | 70.40  | 2.89     |
>
> Table 7b:  RL Reward Ablations
> | Ablation                                 | Recall | Searches |
> | ---------------------------------------- | ------ | -------- |
> | Qwen2.5-7B-Instruct w/ Query Opt         | 66.8   | 2.57     |
> | FrugalRAG-Explore w/ Query Opt           | 76.0   | 5.76     |
> | FrugalRAG w/ Query Opt                   | 68.3   | 2.45     |
> | FrugalRAG w/ Absolute Diff Reward        | 76.0   | 6.00     |
> | FrugalRAG w/o Late Penalty               | 70.5   | 3.144    |
>
>
> (i) **FrugalRAG w/ Absolute Diff Reward:** Here, instead of a logarithmic reward, we use the absolute difference, $\Delta = |h_{\text{term}} - h^{*}|$. We find that it is ineffective in decreasing the total number of searches.
>
> (ii) **FrugalRAG w/ Query Opt:** tunes both the query generation using recall reward and stopping (original FrugalRAG). We find that optimizing both simultaneously is harder and typically decreases overall performance.
>
> (iii) **Stage-2 without Stage-1 (w/o SFT):** directly RL finetunes the base policy Qwen2.5-7B-Instruct without stage 1 finetuning. Our experiment clearly demonstrates that stage 1 finetuning has a positive effect on the overall recall (66 vs 70).
>
> (iv) **FrugalRAG w/o Late Penalty:** Here, we remove the late stop penalty from Equation 1 in the main paper. As expected, the number of searches does not reduce compared to original FrugalRAG (2.89 vs 3.144) achieving similar overall recall.
>
> (v) **Other base model:** To assess whether FrugalRAG generalizes beyond the Qwen2.5-7B-Instruct base model, we conduct additional experiments using Llama3.1-8B-Instruct. In this setting, we reuse the Stage-1 training data previously generated by Qwen2.5-7B-Instruct during our initial experiments and finetune Llama3.1-8B-Instruct.
>
> Table 8: The results on Llama3.1-8B-Instruct show that FrugalRAG is model agnostic.
>
> | Base Model           | Dataset  | Recall | Searches |
> | -------------------- | -------- | ------ | -------- |
> | FrugalRAG (Qwen2.5-7B-Instruct)  | HotPotQA | 70.40  | 2.89     |
> | FrugalRAG (Llama3.1-8B-Instruct) | HotPotQA | 70.00  | 2.95     |
> | FrugalRAG (Qwen2.5-7B-Instruct)  | 2Wiki    | 60.40  | 3.62     |
> | FrugalRAG (Llama3.1-8B-Instruct) | 2Wiki    | 60.41  | 3.79     |
> | FrugalRAG (Qwen2.5-7B-Instruct)  | MuSiQue  | 53.70  | 3.74     |
> | FrugalRAG (Llama3.1-8B-Instruct) | MuSiQue  | 53.00  | 3.96     |
>
> **Code release.** We will release the full codebase and models after acceptance.
>
> **Evaluation Metrics.** Thank you for your comment. Standard string-based metrics such as EM, F1, and Match Score are unreliable for evaluating answer quality in multi-hop QA. These metrics are highly sensitive to form (short vs long answer) and do not reliably reflect semantic correctness. Following prior works list R1-Searcher, BrowseComp-Plus, we conduct all our evaluation using MBE. To illustrate why string matching metrics fail we provide two concrete examples:
>
> - If the gold answer is "Advanced Micro Devices" and the model outputs "AMD", the factual answer is correct, yet EM and F1 is 0 despite clear semantic equivalence.
>
> - For a question such as "Who was born first, Trump or Obama?"", an answer like "Among Trump and Obama, Obama is older" receives a perfect Match Score and a high F1, even though the answer is incorrect. This inflates the metric even when the model response is incorrect.
>
> Because of these issues, EM/F1 do not reliably measure accuracy. The datasets we use also do not involve stylistic variation because answers are short, factual spans. Therefore, MBE is appropriate and does not introduce subjective stylistic bias. We use the standardized evaluation prompt from BrowseComp-Plus to ensure consistent judgment across all models.

---

> ### Author Response · Authors · 2025-11-21
> **Rebuttal Part 3**
>
> **Evaluation Metrics. (continued)** Retriever Metrics: In multi-hop QA, retrieval differs from standard IR because of sequential queries rather than producing a single ranked list. Thus IR metrics such as precision@K, MAP/MRR, or other ranking-based measures are not directly applicable. The key metric in multi-hop retrieval is recall over all the steps, since missing any required evidence makes the reasoning worse. Prior work like CoRAG and Search-R1 only focus on answer-level metrics. That said, to strengthen our evaluation, we now report document-level overall precision and F1 in addition to recall. These metrics capture retrieval quality without relying on ranking assumptions that do not hold in the multi-hop setting. Metrics like precision@K or MRR are standard for standalone retrievers, which is not the focus of our work.
>
> Table 9: HotPotQA Retrieval Metrics
> | Model                  | Rec     | Prec  | F1    | Searches |
> | ---------------------- | ------- | ----- | ----- | -------- |
> | O2 Searcher        | 50.10   | 22.29 | 30.85 | 1.77     |
> | SimpleDeepSearcher | 64.80   | 23.43 | 34.42 | 2.75     |
> | R1 Searcher       | 69.10   | 24.25 | 35.90 | 2.22     |
> | Search-R1          | 48.20   | 20.24 | 28.51 | 1.28     |
> | CoRAG              | 64.30   | 24.03 | 34.99 | 4.00     |
> | **FrugalRAG (ours)**   | 70.40   | 26.81 | 38.83 | 2.89     |
>
> Table 10: 2Wiki Retrieval Metrics
> | Model                  | Rec | Prec  | F1    | Searches |
> | ---------------------- | ------- | ----- | ----- | -------- |
> | O2 Searcher        | 55.20   | 21.99 | 26.42 | 2.42     |
> | SimpleDeepSearcher | 60.50   | 17.97 | 24.38 | 3.64     |
> | R1 Searcher        | 60.40   | 21.60 | 29.29 | 2.36     |
> | Search-R1          | 47.70   | 15.89 | 20.78 | 1.89     |
> | CoRAG              | 65.40   | 22.79 | 30.67 | 4.00     |
> | **FrugalRAG (ours)**   | 59.90   | 15.45 | 23.09 | 3.03     |
>
> Table 11: MuSiQue Retrieval Metrics
> | Model                  | Rec | Prec  | F1    | Searches |
> | ---------------------- | ------- | ----- | ----- | -------- |
> | O2 Searcher        | 37.00   | 14.25 | 21.37 | 1.95     |
> | SimpleDeepSearcher | 50.40   | 14.79 | 22.87 | 2.86     |
> | R1 Searcher        | 57.70   | 15.77 | 24.77 | 2.31     |
> | Search-R1          | 38.10   | 12.65 | 19.86 | 1.36     |
> | CoRAG            | 54.00   | 16.71 | 25.97 | 4.00     |
> | **FrugalRAG (ours)**   | 52.60   | 15.13 | 23.96 | 3.30     |
>
> **Stability of RL training.** The RL training is stable across seeds. We conduct three independent runs using 1000 examples from the HotPotQA datasets and test on all three dataset. The recall on HotPot evaluation is $69.60\pm1.21$ using $2.74\pm0.16$ searches on average. On 2WikiMultiHopQA and MuSiQue we achieve a recall of $59.2\pm1.11$ (searches - $3.60\pm0.11$) and $53.10\pm1.4$ (searches - $3.42\pm0.08$) respectively. We will include this experiment in the revised version of the manuscript.
>
> **Safety in high stakes settings.** Thank you for your question. Our training framework allows us to control the strictness of the stopping criterion. Specifically, $R_{\text{max}}$ can be increased or decreased to control the severity of penalizing the early stop condition in our reward design which can enable us to prevent stopping too early.

---

### Official Review · Reviewer_GAQh · 2025-11-01

**Soundness:** 3
**Presentation:** 3
**Contribution:** 3
**Rating:** 6
**Confidence:** 3

**Summary:**

The paper proposes FrugalRAG, a two‑stage finetuning framework that aims to optimize both accuracy and test‑time retrieval cost for multi‑hop QA in RAG systems. Stage 1 (“Explore”): supervised finetuning of a reasoner to maximize evidence coverage by generating diverse ReAct‑style thought/action/search tuples; dataset rollouts are constructed by greedily selecting the candidate query at each hop that maximizes recall against ground‑truth evidence. FINISH is included in only ~10% of traces to keep it in distribution while emphasizing exploration.  Stage 2 (RL): GRPO is used to learn when to stop searching. The reward penalizes stopping before or after an “optimal” hop count h^\* defined by when further retrieval no longer improves a metric c (thresholded by \tau); an additional format reward encourages valid tool calls. The optimization trains the reasoner only; the final answer generator is fixed.

**Strengths:**

1. Simple yet effective idea: learn when to stop searching rather than always searching more.

2. Strong empirical results with small data (1k); especially on HotPotQA under standardized E5/FlashRAG settings

3. Adaptive compute validated quantitatively and qualitatively (fewer searches on easy, more on hard).

4. Generalization to BrowseCompPlus with more hops than seen in training.

5. Can pair with different retrievers and answerers; demonstration with CoRAG

**Weaknesses:**

I don't identify major weakness for this work, but have a few questions:

1. Reward design depends on oracle‑like signals (evidence recall) and somewhat opaque hyperparameters

2. No baseline learned/heuristic stoppers, so it’s unclear how much RL beats simpler termination rules.

3. Efficiency claims would benefit from runtime/FLOPs and ablation on budget B and sample count v.

4. Practicality depends on availability of ground‑truth evidence for training.

**Questions:**

See above.

---

> ### Author Response · Authors · 2025-11-21
> **Rebuttal Part 1**
>
> We sincerely thank you for your time and effort in reviewing our work. Below, we address all of the concerns raised.
>
> **Evidence Recall Signal.** Our motivation stems from open ended QA on private data or rapidly changing data (e.g., new events). Instead of training a model to optimize final answer accuracy (where many of the answers may be inferred from parametric memory), we train the  model to search for the right documents efficiently, not just the final answer. We agree that this may be limited availability, that's why we show that only 1000 of such dense rewards are needed. In another ablation (Table 10), we show how our algorithm works even if 500 examples are given. This number can be curated easily by organizations who have a specific RAG usecase.
>
> **Hyperparameters.** The only hyperparameters in our reward are $\alpha$ and $R_{\text{max}}$ which are both set to a constant value 2 across all datasets. We will clarify this in the revised version of our manuscript.
>
> **Heuristic Stoppers.** Thank you for the suggestion. To address this concern, we compared FrugalRAG against heuristic stopping rules that do not require training a separate model, ensuring the comparison isolates the effect of our RL-based termination policy. In addition, other heuristic based stopping mechanisms such as ReAct and IRCOT. are already provided as baselines in Table 2 of the main paper.
>
> * Stopping Heuristic 1: We estimated the peak-recall retrieval depth from the training data using the FrugalRAG-Explore policy (HotPotQA: 2.96, 2Wiki: 1.93, MuSiQue: 4.96) and rounded these to fixed stopping points.
> * Stopping Heuristic 2: In the second experiment, we used the average number of retrieval steps required using the ground truth evidence (HotPotQA: 2.0, 2Wiki: 2.40, MuSiQue: 2.32) and rounded these to the nearest integer. The Table below summarizes our results.
>
> Table 1: FrugalRAG compared with fixed heuristic determined stopping criterion.
>
> | Method (HotPotQA)      | Model Variant                       | Recall      | Searches    |
> | ------------ | ----------------------------------- | -------- | -------- |
> | Heuristic 1 | Qwen2.5-7B-Instruct         | 67.7  | 3       |
> |Heuristic 1 | FrugalRAG-Explore   | 70.90        | 3
> | Heuristic 2  | FrugalRAG-Explore            | 67.9     | 2       |
> | Ours  | **Original FrugalRAG**                   | 70.4 | 2.89 |
>
> | Method (2Wiki) | Model Variant                  | Recall | Searches     |
> | ------- | ------------------------------ | ------------ | -------- |
> | Heuristic 1, 2   | Qwen2.5-7B-Instruct         | 52.0         | 2        |
> |  Heuristic 1, 2       | FrugalRAG-Explore    | 55.4         | 2        |
> |  Ours       | **Original FrugalRAG**              | 58.80    | 3.03 |
>
>
> | Method (MuSiQue) | Model Variant                | Recall | Searches     |
> | ------- | ---------------------------- | ------------ | -------- |
> | Heuristic 1 | Qwen2.5-7B-Instruct         | 51.8         | 5        |
> |  Heuristic 1       | FrugalRAG-Explore | 54.5         | 5        |
> | Heuristic 2        | FrugalRAG-Explore      | 43.8         | 2        |
> |  Ours       | **Original FrugalRAG**            | 53.30    | 3.30 |
>
> Across HotpotQA, 2WikiMultiHopQA, and MuSiQue, the baselines consistently underperform our FrugalRAG approach. This shows that our method captures instance-specific termination behavior that simple stopping heuristics cannot. Notably, the benefits are largest on harder datasets like MuSiQue and 2WikiMultiHopQA, where the optimal number of hops varies more substantially across examples.

---

> ### Author Response · Authors · 2025-11-21
> **Rebuttal Part 2**
>
> **Efficiency claims.** Thank you for your suggestion. We have conducted additional ablations on efficiency to support our claims:
>
> **Additional Metrics.** We computed FLOPs, tokens processed, and latency. As shown in the tables below, FrugalRAG is competitive on compute-normalized metrics like tokens, FLOPs, and searches across datasets. We note that in practical deployments monetary cost is typically dominated by the number of tool calls (searches) and LLM tokens, on which FrugalRAG is directly comparable to, and often more efficient than, the baselines.
>
> Table 2: Efficiency Comparison on HotPotQA
>
> | **Method**             | **Avg Tokens** | **FLOPs**              | **Latency (s)** | **Search Count** | **Recall (%)** |
> | ---------------------- | -------------- | ---------------------- | --------------- | ---------------- | -------------- |
> | **O2 Searcher-3B**        | 2,937          | $1.79 \times 10^{13}$  | 0.1050          | 1.77             | 50.10          |
> | **SimpleDeepSearcher-7B** | 9,657          | $1.31 \times 10^{14}$  | 0.6571          | 2.75             | 64.80          |
> | **R1 Searcher-7B**        | 4,458          | $6.17 \times 10^{13}$  | 0.1335          | 2.22             | 69.10          |
> | **Search-R1-7B**          | 2,119          | $2.99 \times 10^{13}$  | 0.0650          | 1.28             | 48.20          |
> | **CoRAG-8B**              | 7,770          | $1.24 \times 10^{14}$  | 0.1486          | 4.00             | 64.30          |
> | **FrugalRAG-7B (ours)**   | 9,138      | $1.27 \times 10^{14}$ | 0.2415          | 2.89             | 70.40          |
>
>
> Table 3: Efficiency Comparison on 2WikiMultiHopQA
> | **Method**             | **Tokens** | **FLOPs**              | **Latency (s)** | **Search Count** | **Recall (%)** |
> | ---------------------- | ---------- | ---------------------- | --------------- | ---------------- | -------------- |
> | **O2 Searcher-3B**        | 4,212      | $2.53 \times 10^{13}$  | 0.1206          | 2.42             | 55.20          |
> | **SimpleDeepSearcher-7B** | 13,433     | $1.80 \times 10^{14}$  | 0.7625          | 3.64             | 60.50          |
> | **R1 Searcher-7B**        | 4,939      | $6.89 \times 10^{13}$  | 0.1374          | 2.36             | 60.40          |
> | **Search-R1-7B**          | 3,456      | $4.96 \times 10^{13}$  | 0.0852          | 1.89             | 47.70          |
> | **CoRAG-8B**              | 8,018      | $1.28 \times 10^{14}$  | 0.1511          | 4.00             | 65.40          |
> | **FrugalRAG-7B (ours)**   | 11279      | $1.45 \times 10^{14}$ | 0.4673          | 3.03             | 59.90          |
>
>
> Table 4: Efficiency Comparison on MuSiQue
> | **Method**             | **Tokens** | **FLOPs**              | **Latency (s)** | **Search Count** | **Recall (%)** |
> | ---------------------- | ---------- | ---------------------- | --------------- | ---------------- | -------------- |
> | **O2 Searcher-3B**        | 2,923      | $1.74 \times 10^{13}$  | 0.1009          | 1.95             | 37.00          |
> | **SimpleDeepSearcher-7B** | 10,027     | $1.35 \times 10^{14}$  | 0.7366          | 2.86             | 50.40          |
> | **R1 Searcher-7B**        | 4,665      | $6.47 \times 10^{13}$  | 0.1327          | 2.31             | 57.70          |
> | **Search-R1-7B**          | 2,212      | $3.14 \times 10^{13}$  | 0.0676          | 1.36             | 38.10          |
> | **CoRAG-8B**              | 7,683      | $1.23 \times 10^{14}$  | 0.1472          | 4.00             | 54.00          |
> | **FrugalRAG-7B (ours)**   | 11914     | $1.66 \times 10^{14}$ | 0.4098          | 3.30             | 52.60          |

---

> ### Author Response · Authors · 2025-11-21
> **Rebuttal Part 3**
>
> **Budget Ablation.** We evaluate a HotPotQA model trained with a budget of 6 and test it under a range of budgets (B = 1–11). As the budget increases, the model adaptively issues more searches, leveraging the additional allowance to retrieve more evidence. Recall improves sharply when moving from very small budgets (B = 1–2) to moderate budgets (B = 3–6), indicating that additional retrievals in this range meaningfully expand the evidence available to the model. Beyond B = 6, the improvements become marginal on HotPotQa, 2WikiMultiHopQA, and MuSiQue, despite the model continuing to issue more searches. On other more complex datasets, additional retrievals may continue to help. Notably, the highest recall is achieved at B = 6, matching the budget used during training. This behavior is expected as the model has learned retrieval strategies tailored to this budget.
>
> Table 5: Budget ablation
> | **Budget (B)** | **Recall (%)** | **Searches** |
> | -------------- | -------------- | ------------ |
> | 1              | 52.7           | 1.00         |
> | 2              | 65.1           | 1.76         |
> | 3              | 67.9           | 2.15         |
> | 4              | 68.6           | 2.41         |
> | 5              | 68.8           | 2.63         |
> | 6              | 70.4           | 2.89         |
> | 7              | 69.3           | 2.93         |
> | 8              | 68.6           | 3.07         |
> | 9              | 69.1           | 3.22         |
> | 10             | 68.8           | 3.30         |
> | 11             | 69.2           | 3.46         |
>
>
> **Sample Count Ablation.** Across all datasets, performance increases substantially from 100 to 500 examples. For example, from 100 to 500 examples, the recall improves from 67.9 to 70.80 on HotpotQA, and the same trend appears in 2Wiki and Musique. This shows that our method quickly learns the core reasoning required for retrieval with only a few hundred supervised examples. Interestingly, the search efficiency stabilizes with more data (with 5000 examples searches drop consistently across the three datasets). This substantiates that with increasing number of examples our method improves the overall performance.
>
>
> Table 6: Sample Count Ablation
> | **Dataset**  | **Metric** | **100** | **500** | **1000** | **5000** |
> | ------------ | ---------- | ------- | ------- | -------- | -------- |
> | **HotpotQA** | Recall     | 67.9    | 70.8    | 70.4     | 69.1     |
> |              | Searches   | 2.873   | 3.243   | 2.89     | 2.36     |
> | **2Wiki**    | Recall     | 58.5    | 59.1    | 60.04    | 59.3     |
> |              | Searches   | 3.54    | 3.79    | 3.623    | 3.01     |
> | **MuSiQue**  | Recall     | 51.6    | 53.6    | 53.7     | 52.4     |
> |              | Searches   | 3.532   | 3.774   | 3.47     | 3.038    |
>
>
> **Availability of Ground Truth.** Yes, we acknowledge that our method depends on the availability of ground truth training data. However, we would like to reiterate that only a small amount of ground truth examples (500-1000) are required to train FrugalRAG which makes it significantly more practical compared to existing work.

---

> > ### Comment · Reviewer_GAQh · 2025-11-21
> >
> > Thank you for your detailed response. Most of my concerns have been solved and I appreciate the great effort of authors in conducting new experiments to address those concerns. I increase my score accordingly.

---

### Author Response · Authors · 2025-12-02
**Rebuttal Summary**

As the discussion period ends, we would like to sincerely thank all reviewers and ACs for their time, effort, and thoughtful engagement in improving the quality of our manuscript.

We are encouraged that we were able to fully address all of reviewer GAQh’s concerns (they had raised their score to 8, see their comment below). Although the other reviewers did not participate further, we believe our rebuttal has comprehensively addressed all open questions and concerns. Below, we summarize the key clarifications and additional experiments.


### **Strengths**
We are pleased that all reviewers consistently acknowledge the strengths of our work:

- **Effective idea, logical design, and important problem.** Reviewers highlighted the clarity and actionability of our method (**r6Km**), its simplicity and effectiveness (**GAQh**), clear problem formulation and logical structure (**2odc**), and the importance of the problem addressed (**9twQ**).

- **Strong quantitative results.** All reviewers (9twQ, r6Km, GAQh, 2odc) praised the *strong empirical performance*, especially given the small number of training examples. They also emphasized notable efficiency gains (**2odc**) and appreciated how our tradeoff metric showcases practical benefits (**r6Km**).

- **Modular, adaptive, and generalizable.** Reviewers GAQh and r6Km noted that our approach is modular across models and retrievers, dynamically uses more retrieval calls for harder questions, and generalizes effectively to BrowseComp-Plus.

### **Additional Experiments**

1. **Efficiency Metrics.**
   We added evaluations of FLOPs, tokens, and latency on HotPotQA, MuSiQue, and 2WikiMultiHopQA, demonstrating **competitive efficiency and recall** relative to strong reasoning-based baselines across all three datasets. Notably, this performance is achieved despite the DsPy prompting overhead introduced by our method. Results are shown in Tables 2, 3, and 4. (Reviewers GAQh, 2odc)

2. **Effect of Scaling.**
   We conducted an additional ablation analyzing the effect of the number of training examples. FrugalRAG remains competitive even with as few as 500 examples. As the training set grows beyond 1000 examples, efficiency continues to improve, while overall recall begins to plateau. Results appear in Table 6. (Reviewers GAQh, r6Km, 9twQ)

3. **Effect of the Budget Hyperparameter.**
   As the retrieval budget increases, the model issues more searches, leveraging the larger allowance to gather additional evidence. Recall improves sharply when moving from very small budgets (B = 1–2) to moderate budgets (B = 3–6), indicating that additional retrievals in this range meaningfully expand the available evidence. Beyond B = 6, improvements become marginal, even though the model continues to issue more searches. We anticipate that on more complex datasets, additional retrievals may yield further gains. Results are shown in Table 5. (Reviewers GAQh, 9twQ)

4. **Comparison Against Heuristic Baselines.**
   We compared FrugalRAG to heuristic methods with static stopping conditions derived from training data. As expected, FrugalRAG’s adaptive strategy consistently outperforms these baselines across Tables 1, 2, and 3. (Reviewer GAQh)

5. **Additional Ablations.**
   To isolate the contribution of each component in our reward formulation, we conducted ablations presented in Tables 7, 8, 13, and 15. We show that FrugalRAG outperforms variants without the late-stop penalty, without Stage–1, with alternative reward designs (e.g., Query Opt, absolute-difference reward), as well as the base model with Query Opt alone. These results confirm that the gains from FrugalRAG are significant and not simply a because of regularization. (Reviewers 2odc, r6Km, 9twQ)

6. **Stability of RL Fine-Tuning Across Seeds.**
   We ran three independent training runs using 1000 examples from HotPotQA and evaluated on all three datasets. The standard deviation across runs remains very low for both recall and the number of searches, demonstrating the stability of our RL fine-tuning procedure. (Reviewer 2odc)

7. **Code release.** We confirm that we will release the full codebase and trained models upon acceptance. (Reviewer 2odc)

> We would like to once again express our sincere gratitude to all reviewers for their thoughtful and constructive feedback. We believe that the additional experiments and clarifications have substantially strengthened our paper and thoroughly addressed the concerns raised. We hope that the final version of our work will meaningfully contribute to the ICLR community.

---

### Meta-Review · Area_Chair_DqQt · 2026-01-02

**Summary:**

This paper introduces FrugalRAG, a two-stage fine-tuning framework to optimize both answer accuracy and retrieval efficiency in multi-hop RAG. The method begins with a supervised exploration stage that trains the model to maximize retrieval coverage by generating diverse search queries. Then, subsequent RL stage teaches the model to adaptively determine when to stop. In experiments, FrugalRAG achieves better efficiency-accuracy tradeoffs on multiple RAG benchmarks with significantly less training data.

**Reviewer Concerns:**

Some reviewers raised concerns on the fairness of the evaluation settings (e.g., retrievers). Some reviewers also raised concerns on limited ablations and analyses on the methods. Two reviewers also raised some concerns on the novelty. During the rebuttals, the authors clarified the fairness of the evaluation settings as well as adding more experiments.

**Reviewer Scores:**

Two reviewers provide 6 and two reviewers provide 4. During rebuttal, only one reviewer participates the discussion and raised the score from 6 to 8. Unfortunately, the other three reviewers did not participate the discussion even though the authors provided very detailed rebuttal with experiments. Based on AC's evaluation on the rebuttals, many of the raised concerns are resolved.

---

### Decision · Program_Chairs · 2026-01-26

Accept (Poster)